# Computational Budget Should Be Considered in Data Selection

**Weilin Wan**[1], **Weizhong Zhang**[2]*, **Cheng Jin**[1]
[1]College of Computer Science and Artificial Intelligence, Fudan University
[2]School of Data Science, Fudan University
`wlwan23@m.fudan.edu.cn`, `{weizhongzhang, jc}@fudan.edu.cn`

## Abstract

Data selection improves computational efficiency by choosing informative subsets of training samples. However, existing methods ignore the compute budget, treating data selection and importance evaluation independently of compute budget constraints. Yet empirical studies show no algorithm can consistently outperform others (or even random selection) across varying budgets. We therefore argue that compute budget must be integral to data-selection strategies, since different budgets impose distinct requirements on data quantity, quality, and distribution for effective training. To this end, we propose a novel Computational budget-Aware Data Selection (CADS) method and naturally formulate it into a bilevel optimization framework, where the inner loop trains the model within the constraints of the computational budget on some selected subset of training data, while the outer loop optimizes data selection based on model evaluation. Our technical contributions lie in addressing two main challenges in solving this bilevel optimization problem: the expensive Hessian matrix estimation for outer-loop gradients and the computational burden of achieving inner-loop optimality during iterations. To solve the first issue, we propose a probabilistic reparameterization strategy and compute the gradient using a Hessian-free policy gradient estimator. To address the second challenge, we transform the inner optimization problem into a penalty term in the outer objective, further discovering that we only need to estimate the minimum of a one-dimensional loss to calculate the gradient, significantly improving efficiency. To accommodate different data selection granularities, we present two complementary CADS variants: an example-level version (CADS-E) offering fine-grained control and a source-level version (CADS-S) aggregating samples into source groups for scalable, efficient selection without sacrificing effectiveness. Extensive experiments show that our method achieves performance gains of up to 14.42% over baselines in vision and language benchmarks. Additionally, CADS achieves a 3-20× speedup compared to conventional bilevel implementations, with acceleration correlating positively with compute budget size.

## 1 Introduction

Model training costs have been increasing rapidly, and as training data accumulates over time, much of it becomes redundant. This makes data selection a crucial approach to reduce computational burden and enhance model performance [52, 59]. Numerous studies have tackled the challenge of selecting informative subsets of training samples, employing diverse criteria to identify the most impactful data for training [2]. Most existing methods fall into two categories. The first category proposes a measurement metric and selects the top-K samples based on the scores derived from this

---

*Corresponding Author.

metric [25, 63, 70]. Some methods further incorporate similarity measures, such as cosine similarity, to ensure diversity among the selected samples [63]. The second category aims to match certain distributions, such as gradients, embeddings, or other indicative features, to better represent the overall dataset [26, 27, 41, 69].

However, despite numerous advances in data selection methods, a critical gap remains: most approaches overlook the computational budget as a key factor shaping the selection process. Empirical studies [18, 22] reveal that optimal model performance depends on balancing training data volume, model scale, and available compute budget, highlighting that data selection is inherently linked to computational constraints. Moreover, recent studies [11, 54, 62] reveal that sophisticated selection strategies often fail to consistently outperform random selection across varied experimental settings. [67] further demonstrate that when computational budgets are explicitly factored in, previously effective data selection approaches rarely remain optimal. *We thus argue that computational budget must be integral to data selection strategies*, as it determines the appropriate quantity, quality, and distribution of training data, making it a first-order design decision rather than a fixed hyperparameter.

Our argument is further supported by neural network learning dynamics. Models rely on different granularities of knowledge at various training stages, reflected in distinct data subsets. Rahaman et al. [50] showed that networks exhibit a spectral bias, learning low-frequency features before higher-frequency ones. Under limited computational budgets, focusing on data rich in low-frequency features can optimize learning, whereas larger budgets allow leveraging diverse, higher-frequency information. We verify these effects empirically in Section 3.

In this paper, we propose a novel Computational Budget-Aware Data Selection (CADS) method, which we naturally formulate within a bilevel optimization framework. In this framework, the inner loop trains the model under the constraints of the computational budget on a selected subset of training data, while the outer loop focuses on optimizing data selection based on the evaluation of the model trained in the inner loop. Our technical contributions target two main challenges: the expensive estimation of the Hessian matrix for gradients in the outer loop and the computational burden associated with achieving optimality in the inner loop during iterations. To solve the first challenge, we introduce a probabilistic reparameterization strategy and utilize a Hessian-free policy gradient estimator to compute the gradient efficiently. For the second challenge, we reformulate the inner optimization problem as a penalty term in the outer objective function. Notably, we find that estimating the minimum of a one-dimensional loss is sufficient for calculating the gradient, significantly enhancing computational efficiency.

To meet varying data selection demands in practice, we implement CADS in two variants addressing different operational requirements: CADS-E operates at example-level granularity for precise control, while CADS-S assigns weights to source groups for improved scalability. Experiments reveal performance improvements of up to 14.42% over baselines across vision and language tasks. CADS delivers 3-20× speedup versus conventional bilevel methods, with acceleration scaling proportionally to compute budget size.

**Summary of contributions:**

- We highlight the crucial role of computational budget in data selection, advocating it as a first-order design factor rather than a fixed hyperparameter, thus addressing a key gap by linking data selection to computational constraints.

- We propose a novel bilevel optimization framework for Computational Budget-Aware Data Selection (CADS), where the inner loop trains the model under budget constraints on a selected data subset, and the outer loop optimizes data selection based on the trained model's evaluation.

- To efficiently solve this bilevel problem, we develop techniques including a probabilistic reparameterization for gradient estimation avoiding expensive Hessian computation, and a reformulation of the inner problem as a penalty term in the outer objective. We further show that estimating a one-dimensional loss minimum suffices for gradient calculation, greatly improving computational efficiency.

- Extensive empirical evaluations demonstrate that our CADS framework achieves superior performance across multiple datasets and training settings, validating the effectiveness of computational budget-aware data selection.

# 2 Related work

## 2.1 Data selection

**Traditional data selection methods and related concepts.** Data selection methods aim to identify subsets of training samples that maximize informativeness or diversity and minimize redundancy [2]. Approaches leveraging submodular optimization and clustering have shown success in various tasks [40, 51, 61]. Coreset selection techniques specifically construct small representative subsets that retain essential properties of the full dataset, often using importance weights or gradient similarity [26, 51]. Learning-based methods integrate selection into training through training dynamics analysis [44, 59], bilevel optimization [27, 74], gradient properties [21, 23], or uncertainty measures [3]. Dataset distillation methods construct synthetic examples to compress large datasets, enabling faster or more efficient training by matching gradient statistics [6, 10, 60, 68]. While highly compact, distilled sets often struggle to match real-data diversity or scale with heterogeneous corpora characteristic of large-scale model training. Curriculum learning [4] dynamically orders or selects data according to difficulty, often progressing from easy to challenging samples [17, 55]. A related body of work uses importance sampling and adaptive reweighting to prioritize data points that contribute most to model improvement or convergence [1, 4, 36], demonstrating efficiency gains especially in large-scale or imbalanced data regimes.

**Data selection for LLM fine-tuning.** Unlike traditional methods, data selection for LLMs often employs simpler heuristic rules due to the unprecedented scale of both models and datasets. Efficient subset selection has become crucial in LLM fine-tuning [2, 39, 42, 49]. LIMA demonstrated strong performance with just 1,000 human-curated examples [72], while automated approaches have emerged to replace manual curation. These include natural language indicators with BlendSearch [5], semantic intention tagging [37], model-based quality filtering [7, 31], instruction difficulty metrics [30], and uncertainty-driven active learning [28]. Recent advances explore gradient-based techniques, including clustered coreset selection [69] and graph methods [71], alongside simple yet effective heuristics like length-based prioritization [70]. Novel approaches leverage small model training trajectories to guide selection for larger models [64] and sparse autoencoders for diversity-driven selection [63]. DEITA provides a unified framework balancing quality, complexity, and diversity [34]. These methods achieve comparable or superior performance to full-dataset training while reducing computational requirements, demonstrating that strategically selected subsets can match or exceed naive scaling.

**The compute-aware gap.** Existing methods typically operate with predefined data budgets without considering computational constraints. Recent studies [11, 54, 62] reveal that sophisticated selection strategies often fail to consistently outperform random selection across varied experimental settings. [67] further demonstrate that when computational budgets are explicitly factored in, previously effective data selection approaches rarely remain optimal. This computational perspective motivates our approach to data selection that explicitly accounts for training budget constraints.

## 2.2 Bilevel optimization

**Algorithmic advances** Early work on bilevel optimization in deep learning focused on implicit differentiation and Hessian–vector products [12, 16, 45] as well as iterative differentiation with truncated back-propagation [38, 53, 73]. These methods scale poorly because they require nested loops and second-order information. Stochastic variants [9, 15, 19, 20, 24] cut per-iteration cost but still depend on Jacobian/Hessian evaluations. Finite-difference estimators [56, 65] remove explicit Hessians yet introduce instability. Most recently, fully first-order, penalty-based formulations [8, 29, 66] recast the inner optimality conditions as constraints, eliminating higher-order derivatives and enabling training with only standard gradients. This advancement enables bilevel algorithms to be applied to increasingly larger-scale problems.

**Practical applications** Bilevel optimization has become a fundamental tool in machine learning, initially enabling hyperparameter optimization by jointly optimizing model parameters and hyperparameters [14]. Key applications include learning-rate and weight-decay tuning [13, 35]. In neural architecture search, DARTS frames architecture parameters as outer variables and weights as inner variables, achieving efficient architecture discovery [33]. Resource-aware tasks benefit from probabilistic bilevel formulations for memory-constrained coreset selection [74], and gradient-matching

methods for learning compact synthetic datasets [60]. Extending to large-scale settings, ScaleBiO applies first-order bilevel data reweighting at the scale of billions of tokens in language modeling, demonstrating improved performance with manageable computational overhead [43]. These developments highlight bilevel optimization's versatility in structuring complex learning problems, enabling efficient parameter tuning, architecture search, and data management, and suggest continued expansion into broader applications.

# 3 An exploratory experiment to illustrate the impact of compute constraints on data choice

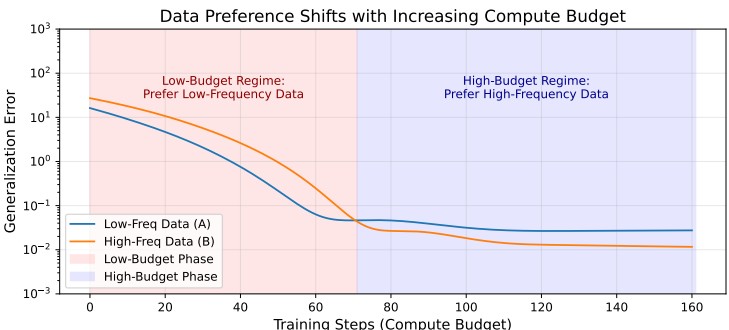

Figure 1: Validation loss of models trained on low- and high-frequency data under varying compute budgets. Two regimes emerge: low-budget favors low-frequency data, while high-budget favors high-frequency data, showing compute's effect on data choice.

The spectral bias of neural networks—favoring learning low-frequency features before higher-frequency ones—has been well documented in prior work [50]. Their findings highlight that models naturally prioritize simpler, low-frequency information early in training. In this section, we build upon this insight to explore the relationship between computational budget and data selection more explicitly. To visualize this connection, we adopt a simple synthetic data generation scheme designed for interpretability.

Specifically, inspired by [50], data are sampled from the function $y = x + \frac{\sin(\pi x)}{\pi x}$ (where the fraction equals 1 when x = 0) with additive Gaussian noise. The low-frequency dataset (Model A) samples exclusively at $\sin(\pi x) = 0$ points, capturing linear patterns. The high-frequency dataset (Model B) samples uniformly across the domain, preserving spectral complexity. Both datasets contain 50,000 samples each to control for size effects, isolating differences to spectral properties alone. We employ a three-layer fully connected network with ReLU activations; architectural details appear in the appendix. Figure 1 presents validation loss trajectories across varying computational budgets. With limited compute, the low-frequency model demonstrates superior generalization. As compute increases, the high-frequency model achieves better performance, confirming the advantage of richer data when resources permit. These findings establish two distinct regimes in data preference determined by computational constraints, necessitating budget-aware data selection methods that dynamically adapt to available compute—a central focus of this work.

# 4 Methods

This section introduces **Compute-Aware Data Selection (CADS)**, a single objective that jointly optimizes (i) model parameters and (ii) both *how much* and *which* data to process under a fixed compute budget. We first formalise compute-constrained selection as a bilevel optimization problem and review a direct policy-gradient baseline. The remainder of the section presents two CADS variants: CADS-E, operating at the example level, and CADS-S, operating at the source level.

## 4.1 Compute-constrained data selection

To address the limitations of traditional data selection methods, we redefine the problem with the computational budget as an explicit constraint. Given a fixed compute limit $C$, we aim to identify a subset of data—along with its size—that maximizes validation accuracy while ensuring that the total number of sample-processing steps does not exceed $C$. This can be formalized as the following bilevel optimization problem:

$$\min_{\boldsymbol{m}} \mathcal{L}_{\text{val}}(\boldsymbol{\theta}_C(\boldsymbol{m})) \quad \text{s.t.} \ \boldsymbol{\theta}_C(\boldsymbol{m}) = \text{Train}(\boldsymbol{m}, C), \tag{1}$$

where $\boldsymbol{m}$ is a binary selection vector indicating which samples to include in training, and $\boldsymbol{\theta}_C$ represents the model parameters derived from training with budget $C$ using the selected subset $\boldsymbol{m}$. A natural approach to this problem is bilevel optimization [12, 13, 16, 35, 38, 45, 53]. To optimize $\boldsymbol{m}$, we compute the gradient of the validation loss with respect to $\boldsymbol{m}$, given by $\nabla_m \mathcal{L}_{\text{val}}(\boldsymbol{\theta}_C(\boldsymbol{m})) = \frac{\partial \mathcal{L}_{\text{val}}}{\partial \boldsymbol{\theta}_C} \cdot \frac{\partial \boldsymbol{\theta}_C}{\partial \boldsymbol{m}}$. Calculating $\frac{\partial \boldsymbol{\theta}_C}{\partial \boldsymbol{m}}$ requires assessing how model parameters vary with $\boldsymbol{m}$, which can be approached in two ways: implicit differentiation at the optimality condition, requiring costly second-order derivatives, or unrolling the training trajectory and backpropagating, which incurs even higher computational costs. Given our emphasis on efficiency within compute constraints, both methods are impractical, especially because the assumption $\nabla_{\boldsymbol{\theta}} \mathcal{L}_{\text{inner}}(\boldsymbol{\theta}, \boldsymbol{m}) = 0$ may not hold. The limited budget $C$ prevents reaching a true local minimum, leaving the inner gradient non-zero and making implicit differentiation unreliable. Inspired by [74], we propose a learnable sampling distribution parameterized by $\boldsymbol{s}$ and optimize $\boldsymbol{s}$ using policy gradients instead of directly optimizing $\boldsymbol{m}$. This approach allows us to navigate the bilevel problem without relying on gradients from the inner optimization under compute constraints. In this formulation, we define $\boldsymbol{m} \in \{0, 1\}^N$ as a binary mask selecting a subset from the training corpus $\mathcal{D}$, with $p(\boldsymbol{m} \mid \boldsymbol{s})$ representing the corresponding sampling distribution. Given a strict computational budget of $C$ forward-backward passes, our objective is to identify the sampler that minimizes the expected validation loss after training for exactly $C$ steps on the masked data:

$$\min_{\boldsymbol{s}} \mathbb{E}_{\boldsymbol{m} \sim p(\boldsymbol{m}|\boldsymbol{s})} \big[ \mathcal{L}_{\text{val}}(\boldsymbol{\theta}_C(\boldsymbol{m})) \big] \quad \text{s.t.} \quad \boldsymbol{\theta}_C(\boldsymbol{m}) = \text{Train}(\boldsymbol{m}, C), \tag{2}$$

where $\boldsymbol{\theta}_C(\boldsymbol{m})$ represents the model parameters obtained by training on the selected subset $\boldsymbol{m}$ with the fixed compute budget $C$. Unlike classical bilevel formulations, where the inner problem is solved to convergence, this formulation explicitly constrains training to $C$ steps, closely matching practical real-world training scenarios.

## 4.2 Bilevel-CADS: policy-gradient baseline

A direct yet costly way to optimize Equation (2) is to treat the selector as a stochastic policy and apply REINFORCE. The sampler is factorised over examples:

$$p(\boldsymbol{m} \mid \boldsymbol{s}) = \prod_{i=1}^{N} s_i^{m_i} (1 - s_i)^{1-m_i} \tag{3}$$

where $s_i \in [0, 1]$ is the inclusion probability for example $i$. The policy gradient of the outer objective is then

$$\nabla_{\boldsymbol{s}} \mathbb{E}_{p(\boldsymbol{m}|\boldsymbol{s})} \big[ \mathcal{L}_{\text{val}}(\boldsymbol{\theta}_C(\boldsymbol{m})) \big] = \mathbb{E}_{p(\boldsymbol{m}|\boldsymbol{s})} \big[ \mathcal{L}_{\text{val}}(\boldsymbol{\theta}_C(\boldsymbol{m})) \nabla_{\boldsymbol{s}} \log p(\boldsymbol{m} \mid \boldsymbol{s}) \big] \tag{4}$$

In practice we approximate the expectation with $K$ Monte-Carlo samples:

$$\nabla_{\boldsymbol{s}} \mathbb{E}_{p(\boldsymbol{m}|\boldsymbol{s})} \big[ \mathcal{L}_{\text{val}}(\boldsymbol{\theta}_C(\boldsymbol{m})) \big] \approx \frac{1}{K} \sum_{k=1}^{K} \mathcal{L}_{\text{val}}(\boldsymbol{\theta}_C(\boldsymbol{m}_k)) \nabla_{\boldsymbol{s}} \log p(\boldsymbol{m}_k \mid \boldsymbol{s}), \quad \boldsymbol{m}_k \sim p(\boldsymbol{m} \mid \boldsymbol{s}) \tag{5}$$

Each gradient update therefore entails training $K$ independent models for $C$ compute units each, yielding a total cost of $K \times C$ per outer step. With even modest values (e.g. $K=5$), this baseline quickly becomes prohibitive and motivates the more efficient relaxations introduced in Sections 4.3.

## 4.3 Penalty-based single-level relaxation

To address the prohibitive computational cost of the policy gradient approach, inspired by [29, 43], we reformulate the bilevel problem as a single-level optimization through a penalty-based relaxation.

Instead of repeatedly training models to exhaustion for each sampled mask, we introduce an auxiliary variable $\boldsymbol{u}$ that acts as a proxy for the fully-trained parameters $\boldsymbol{\theta}_C$ and penalise the discrepancy between the training losses of $\boldsymbol{\theta}$ and $\boldsymbol{u}$:

$$\min_{\boldsymbol{s},\boldsymbol{\theta}} \mathcal{L}^{\alpha}_{\text{penalty}}(\boldsymbol{\theta},\boldsymbol{s}) \triangleq \mathbb{E}_{p(\boldsymbol{m}|\boldsymbol{s})}\Big[\mathcal{L}_{\text{val}}(\boldsymbol{\theta}) + \alpha\big(\mathcal{L}_{\text{trn}}(\boldsymbol{\theta},\boldsymbol{m}) - \mathcal{L}_{\text{trn}}(\boldsymbol{u},\boldsymbol{m})\big)^2\Big] \tag{6}$$

with

$$\boldsymbol{u} = \text{Train}(\boldsymbol{\theta}_0,\boldsymbol{m};K), \quad K = \frac{C}{|\boldsymbol{m}|}.$$

In this formulation, $\mathcal{L}_{\text{trn}}(\boldsymbol{u},\boldsymbol{m})$ serves as a target training loss that reflects the performance of a model trained for a limited number of steps $K$ under the given compute budget $C$.

**Technical difficulty in solving problem (6).** Unlike classical bilevel settings where the inner problem reaches (or approximates) optimality, here the target loss corresponds to an intermediate, compute-constrained solution that generally does not satisfy stationary conditions (i.e., non-zero gradients). This makes direct optimization challenging: naively updating the auxiliary variable $\boldsymbol{u}$ jointly with $\boldsymbol{\theta}$ may violate the compute budget constraint or require costly iterative inner optimization. To avoid the need for explicitly unrolling $K$ training steps for each candidate $\boldsymbol{m}$, we approximate $\mathcal{L}_{\text{trn}}(\boldsymbol{u},\boldsymbol{m})$ with a scale-dependent surrogate $l(|\boldsymbol{m}|)$, which can be efficiently estimated. In practice, $l(|\boldsymbol{m}|)$ is found to be well-approximated by a log-linear function of the subset size $|\boldsymbol{m}|$, capturing how the achievable training loss scales with computational budget. We fit this log-space interpolation using training loss measurements collected at multiple subset sizes, balancing data efficiency and approximation quality. A detailed discussion of the surrogate $l(\cdot)$ is provided in the appendix. Subsequently, substituting the approximation yields the single-level objective:

$$\mathcal{L}^{\alpha}_{\text{CADS}}(\boldsymbol{\theta},\boldsymbol{s}) = \mathbb{E}_{p(\boldsymbol{m}|\boldsymbol{s})}\Big[\mathcal{L}_{\text{val}}(\boldsymbol{\theta}) + \alpha\big(\mathcal{L}_{\text{trn}}(\boldsymbol{\theta},\boldsymbol{m}) - l(|\boldsymbol{m}|)\big)^2\Big]. \tag{7}$$

Finally, by approximating the penalty objective $\mathcal{L}^{\alpha}_{\text{penalty}}$ (6) with the CADS objective $\mathcal{L}^{\alpha}_{\text{CADS}}$ (7), we can efficiently solve the budget-constrained minimization in Problem (6).

**Optimization.** Stochastic estimates of the following gradients are obtained with a single forward/backward pass per sampled subset:

$$\nabla_{\boldsymbol{\theta}} \mathcal{L}^{\alpha}_{\text{CADS}}(\boldsymbol{\theta},\boldsymbol{s}) = \mathbb{E}_{p(\boldsymbol{m}|\boldsymbol{s})}\big[\nabla_{\boldsymbol{\theta}}\mathcal{L}_{\text{val}} + 2\alpha\big(\mathcal{L}_{\text{trn}} - l(|\boldsymbol{m}|)\big)\nabla_{\boldsymbol{\theta}}\mathcal{L}_{\text{trn}}\big], \tag{8}$$

$$\nabla_{\boldsymbol{s}} \mathcal{L}^{\alpha}_{\text{CADS}}(\boldsymbol{\theta},\boldsymbol{s}) = \mathbb{E}_{p(\boldsymbol{m}|\boldsymbol{s})}\big[\big(\mathcal{L}_{\text{val}} + \alpha(\mathcal{L}_{\text{trn}} - l(|\boldsymbol{m}|))^2\big)\nabla_{\boldsymbol{s}}\log p(\boldsymbol{m}\mid\boldsymbol{s})\big]. \tag{9}$$

Any first-order optimizer (e.g. Adam) can then update $(\boldsymbol{\theta},\boldsymbol{s})$ jointly.

### 4.4 CADS-E: example-level selection

In the example-level approach, CADS learns to select individual examples by assigning a parameter $s_i \in [0,1]$ to each example $i$ in the training corpus. We model the sampling distribution $p(\boldsymbol{m}|\boldsymbol{s})$ as independent Bernoulli variables for each example as the same distribution defined in (3). The gradient of $\log p(\boldsymbol{m}|\boldsymbol{s})$ with respect to $s_i$ is straightforward:

$$\frac{\partial}{\partial s_i}\log p(\boldsymbol{m}|\boldsymbol{s}) = \frac{m_i}{s_i} - \frac{1-m_i}{1-s_i}. \tag{10}$$

During optimization, each subset $m_i$ represents a binary mask over the training examples. Algorithm 1 provides the pseudocode implementation.

### 4.5 CADS-S: source-level selection

In the source-level variant, instead of parameterizing selection at the level of individual samples, we aggregates samples into broader source groups and assigns weights at this higher level. Specially, the dataset is partitioned into $n$ sources $\{\mathcal{D}_i\}_{i=1}^{n}$, and CADS learns a sampling ratio vector $\boldsymbol{r} \in [0,1]^n$. We let $p(\boldsymbol{r}\mid\boldsymbol{s})$ be an independent truncated Gaussian for each dimension:

$$p(\boldsymbol{r}\mid\boldsymbol{s}) = \prod_{j=1}^{n} \frac{\phi\big((r^j - s^j)/\sigma\big)}{\sigma[\Phi\big((1-s^j)/\sigma\big) - \Phi\big((-s^j)/\sigma\big)]}, \tag{11}$$

| **Algorithm 1** CADS-E (example-level) | **Algorithm 2** CADS-S (source-level) |
|---|---|
| **Require:** budget $C$, initial $\boldsymbol{\theta}_0, \boldsymbol{s}_0$, subset count $K$ | **Require:** budget $C$, sources $\{\mathcal{D}_i\}$, initial $\boldsymbol{\theta}_0, \boldsymbol{s}_0$, subset count $K$, variance $\sigma$ |
| 1: **while** not converged **do** | 1: **while** not converged **do** |
| 2:  Sample $K$ subsets $\{\boldsymbol{m}_k\} \sim p(\boldsymbol{m}\,|\,\boldsymbol{s})$ | 2:  Sample $K$ ratio vectors $\{\boldsymbol{r}_k\} \sim p(\boldsymbol{r}\,|\,\boldsymbol{s})$ |
| 3:  **for** each $\boldsymbol{m}_k$ **do** | 3:  **for** each $\boldsymbol{r}_k$ **do** |
| 4:    Compute $\mathcal{L}_{\text{trn}}(\boldsymbol{\theta}, \boldsymbol{m}_k)$ | 4:    Form $\mathcal{D}_{\boldsymbol{r}_k}$ by subsampling sources |
| 5:    Compute $\mathcal{L}_{\text{val}}(\boldsymbol{\theta})$ | 5:    Compute $\mathcal{L}_{\text{trn}}(\boldsymbol{\theta}, \mathcal{D}_{\boldsymbol{r}_k})$ |
| 6:    Compute approximate loss $l(|\boldsymbol{m}_k|)$ | 6:    Compute $\mathcal{L}_{\text{val}}(\boldsymbol{\theta})$ |
| 7: | 7:    Compute approximate loss $l(|\mathcal{D}_{\boldsymbol{r}_k}|)$ |
| 8:  **end for** | 8:  **end for** |
| 9:  Estimate $\mathcal{L}_{\text{CADS}}$ using (7) | 9:  Estimate $\mathcal{L}_{\text{CADS}}$ using (7) |
| 10:  Compute gradients $\nabla_{\boldsymbol{\theta}}\mathcal{L}$ and $\nabla_{\boldsymbol{s}}\mathcal{L}$ | 10:  Compute gradients $\nabla_{\boldsymbol{\theta}}\mathcal{L}$ and $\nabla_{\boldsymbol{s}}\mathcal{L}$ |
| 11:  Update $\boldsymbol{\theta} \leftarrow \boldsymbol{\theta} - \eta_{\boldsymbol{\theta}}\nabla_{\boldsymbol{\theta}}\mathcal{L}$ | 11:  Update $\boldsymbol{\theta} \leftarrow \boldsymbol{\theta} - \eta_{\boldsymbol{\theta}}\nabla_{\boldsymbol{\theta}}\mathcal{L}$ |
| 12:  Update $\boldsymbol{s} \leftarrow \boldsymbol{s} - \eta_{\boldsymbol{s}}\nabla_{\boldsymbol{s}}\mathcal{L}$ | 12:  Update $\boldsymbol{s} \leftarrow \boldsymbol{s} - \eta_{\boldsymbol{s}}\nabla_{\boldsymbol{s}}\mathcal{L}$ |
| 13: | 13:  Update variance $\sigma \leftarrow 0.99\,\sigma$ |
| 14: **end while** | 14: **end while** |
| 15: **return** $\boldsymbol{s}^*, \boldsymbol{\theta}^*$ | 15: **return** $\boldsymbol{s}^*, \boldsymbol{\theta}^*$ |

where $\phi$ and $\Phi$ denote the standard normal PDF and CDF. This is a truncated Gaussian centered at $s^j$ with scale $\sigma$. We truncate to $[0, 1]$ to obtain valid sampling values. The denominator normalizes the distribution, ensuring the integral equals 1 after truncation. To stabilize optimization, we anneal $\sigma$ using $\sigma_t = 0.99^t \sigma_0$. This gradually concentrates probability mass around $s^j$, leading to consistent sampling results. We visualize the distribution under different $\boldsymbol{s}$ and $\sigma$ values in the appendix. Algorithm 2 provides pseudo-code of CADS-S.

## 5 Experiments

In this section, we present a series of experiments aimed at rigorously evaluating the performance and robustness of our proposed methods across diverse settings and conditions. Detailed experimental settings and implementation details are provided in the appendix.

### 5.1 Small-scale validation

Our goal is to verify that, under a fixed compute budget of 20,000 sample usages (20 full-epoch trainings on a 1,000-sample MNIST subset), Bilevel-CADS and CADS-E can identify near-optimal training subsets. Specifically, we evaluate four approaches across four initial subset sizes spanning from small to large fractions of the total 1,000 samples (200 to 800 samples); experiments under other compute budgets are presented in the appendix. We compare:

- **Random selection:** directly train and test using randomly chosen subsets at these sizes.
- **PBCS [74]:** selects optimal subsets matching the target sizes following original paper settings, then trains and evaluates models.
- **Bilevel-CADS and CADS-E:** start with initial subsets of similar sizes, then execute their respective algorithms to refine and obtain final subsets for model training.

As shown in Table 1, Bilevel-CADS consistently achieves the highest accuracy across all initial subset sizes, significantly outperforming random selection and other methods. Meanwhile, Figure 2 visualizes how the Bilevel-CADS optimization process converges to a stable subset size near 500 samples regardless of the initial subset size, indicating robustness and effective subset refinement driven by the compute budget constraint.

### 5.2 Performance with heterogeneous data sources

**Small-scale experiments on CIFAR-10.** To validate the effectiveness of CADS-S, we conducted experiments on the CIFAR-10 dataset. Specifically, we reserved 10% of the training set (5,000

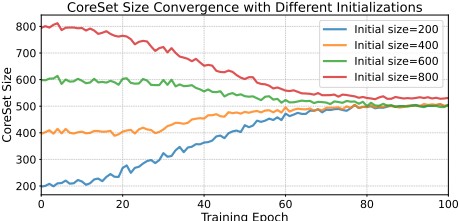

Figure 2: Subset size evolution during optimization with Bilevel-CADS starting from different initial subset sizes.

Table 1: Accuracy (%) comparison of different data selection methods with various initial subset sizes on MNIST.

| Method | Initial subset size | | | | Average |
|---|---|---|---|---|---|
| | 200 | 400 | 600 | 800 | |
| Random | 88.05 | 90.62 | 90.75 | 89.91 | 89.83 |
| PBCS | 89.81 | 92.14 | 92.64 | 92.98 | 91.89 |
| CADS-E | 91.48 | 92.28 | 92.34 | 92.90 | 92.25 |
| Bilevel-CADS | **92.44** | **92.90** | 92.85 | **93.09** | **92.80** |

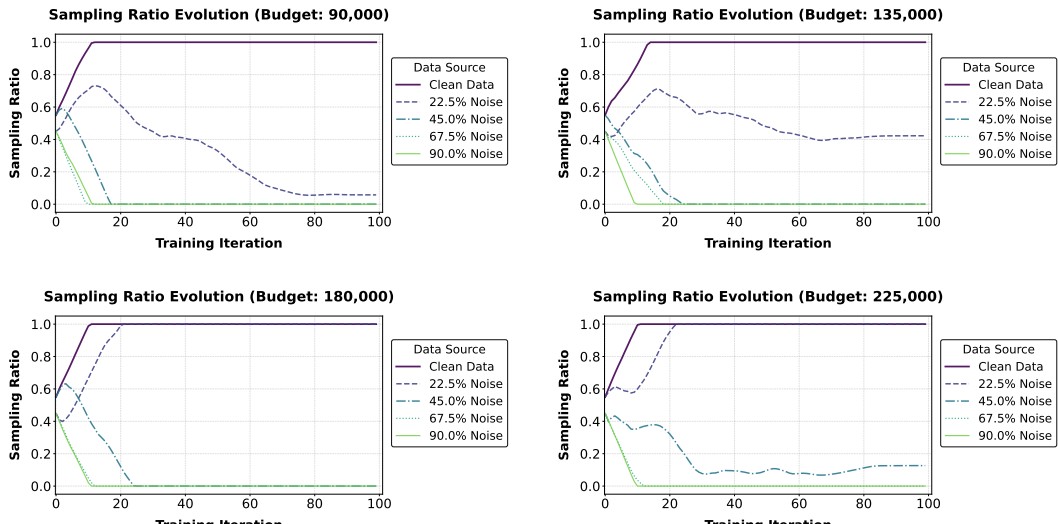

Figure 3: Evolution of the sampling ratio over training iterations under different compute budgets: 90,000, 135,000, 180,000, and 225,000 samples respectively.

samples) as a validation set and partitioned the remaining 45,000 samples into five equal groups of 9,000 each, simulating heterogeneous data sources with noise levels of 0%, 22.5%, 45.0%, 67.5%, and 90.0%, respectively. Our objective was to assess how CADS-S performs compared to training on the single best data source, and the full dataset. The compute budget was varied between 90,000 and 225,000 sample usages.

**Key Insights from Fig. 3:** It clearly illustrates that, under lower compute budgets, our algorithm predominantly favors clean data, assigning near-zero weights to the noisy data sources. However, with an increase in the compute budget, the algorithm gradually elevates the weights of data sources with lower noise levels.

At a compute budget of 225,000 samples, the algorithm incorporates both the clean data and data from the source with 22.5% noise, while also selecting a negligible amount (<0.1) from the source with 45% noise. This phenomenon highlights that as the compute budget expands, even data containing some noise can retain significant value in the training process.

In stark contrast, data sources with noise levels greater than 60% are consistently excluded from selection, regardless of the compute budget in play. These findings suggest that as the compute budget grows, the algorithm becomes increasingly adept at leveraging noisy data, emphasizing the importance of diverse data sources in enhancing model performance.

**Scaling to larger datasets** To further evaluate the scalability and effectiveness of our approach, we conduct experiments on the DomainNet dataset [47], which contains over 0.5 million images across six diverse visual domains: Real, Sketch, Clipart, Painting, Infograph, and Quickdraw. Our setup

Table 2: Accuracy (%) comparison of different data selection methods with various compute budget.

| Method | Compute budget | | | | Average |
|--------|---------|---------|---------|---------|---------|
| | 90,000 | 135,000 | 180,000 | 225,000 | |
| Best-source | 56.31 | 61.59 | 63.72 | 65.08 | 61.65 |
| Full-dataset | 44.34 | 49.80 | 53.34 | 57.03 | 51.13 |
| CADS-S | **57.05** | **62.07** | **65.96** | **67.22** | **63.08** |

Table 3: Accuracy (%) comparison of different data selection methods with various initial subset sizes on DomainNet.

| Method | Initial sampling ratio (%) | | | | Average |
|--------|------|------|------|------|---------|
| | 20 | 40 | 60 | 80 | |
| Uniform-sampling | 29.73 | 37.48 | 39.70 | 37.96 | 36.22 |
| CADS-S | **44.15** | **44.22** | **44.23** | **44.49** | **42.27** |

uses all domains as data sources; however, we treat the Real domain as a high-quality data source and only include a subset of 10,000 samples from it to simulate limited access to the most reliable data. The other five domains are used in full. The total compute budget allocated for training corresponds approximately to training on the full dataset for 10 epochs. Table 3 reports the test accuracy of different data selection methods under varying initial sampling ratios. Our proposed CADS-S method consistently improves over uniform sampling across all ratios. Figure 4 illustrates the evolution of sampling ratios during the optimization process, demonstrating how CADS-S dynamically adjusts data source importance over epochs to better allocate the computational budget.

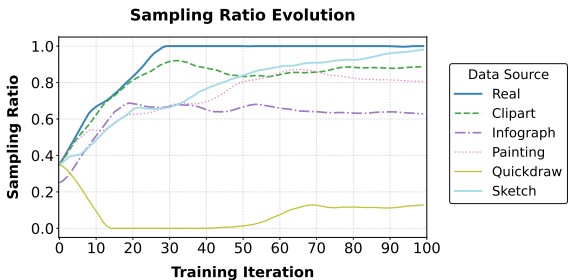

Figure 4: Sampling ratio evolution on DomainNet

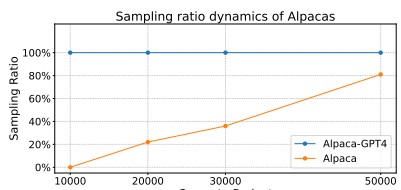

Figure 5: Sampling ratio dynamics of Alpaca-GPT4 and Alpaca datasets under different compute budgets optimized by CADS-S.

## 5.3 Performance on instruction tuning tasks

**Small-scale experiments.** We evaluated our CADS-S approach on instruction fine-tuning tasks using the GPT-2 model [48]. The dataset consists of two heterogeneous data sources: (i) **Alpaca-GPT4** [46], a high-quality dataset from GPT-4-generated instructions, from which we sample 1,000 examples to simulate the rarity of premium data sources in practical scenarios; (ii) **Alpaca** [57], a larger dataset containing 9,000 examples, representing standard quality data. The total compute budget varies within the range $[1 \times 10^4, 5 \times 10^4]$ sample usages. Other settings remain consistent with those described in Section 5.2. As shown in Tab 4, CADS-S consistently outperforms baseline methods, across all compute budgets studied. This validates the proposed method's ability to allocate computational resources between heterogeneous data sources to optimize model performance.

Table 4: Perplexity comparison of different data selection methods with various compute budget.

| Method | Compute budget | | | |
|--------|------|------|------|------|
| | 1e+4 | 2e+4 | 3e+4 | 5e+4 |
| Best-source | 8.01 | 8.02 | 8.21 | 8.87 |
| Full-dataset | 8.26 | 8.01 | 7.90 | 7.77 |
| CADS-S | **8.01** | **7.92** | **7.85** | **7.77** |

Table 5: Perplexity comparison of different data selection methods with various initial subset sizes on additional datasets.

| Method | Initial sampling ratio (%) | | | | Average |
|--------|------|------|------|------|---------|
| | 20 | 40 | 60 | 80 | |
| Uniform-sampling | 7.90 | 7.52 | 7.43 | 7.39 | 7.56 |
| CADS-S | **6.97** | **6.95** | **6.97** | **6.90** | **6.95** |

**Scaling to additional datasets.** We further evaluated CADS across 13 instruction-following fine-tuning datasets, sampling 10,000 examples from each, including AlpacaGPT4 [46], SlimOrca [32], Alpaca [57], GPTeacher [58], and multilingual Alpaca variants, totaling around 130,000 samples.

Detailed dataset setups and preprocessing are provided in the appendix. As shown in Tab 5, our method consistently outperforms baseline approaches in perplexity across different initial sampling ratios, demonstrating its robustness and wide applicability.

## 5.4 Computational Cost Analysis of Established Baselines

To provide a clear comparison of computational efficiency, we establish a unified framework parameterized by:

- **C**: The total compute budget required for a single, standard training run on the full dataset. This serves as our baseline unit of cost.
- **N**: The total number of samples in the full dataset.
- **T**: The number of training epochs, defined as $T = C/N$.

All computational overheads are measured relative to the baseline cost **C** of training on the full dataset.

**A Note on Pre-training Cost:** For methods that require a pre-trained model to compute sample-wise statistics (e.g., gradients, forgetting events), we assume the cost of obtaining this model is **0.2C**. This represents a standard approximation for the early-stage training necessary to yield meaningful statistics, without requiring full convergence.

Table 6: Computational Cost Analysis of Established Baselines.

| Method | Total Computational Overhead |
|---|---|
| Random Selection | C (no additional cost) |
| Forgetting Scores | 1.2C (0.2C for pre-training and forgetting computation + C for final training) |
| GraNd | 1.2C (0.2C for pre-training and gradient computation + C for final training) |
| EL2N | 1.2C (0.2C for pre-training and error computation + C for final training) |
| CADS (Ours) | $(5/A + 2\gamma)$C where A is amortization factor |
| Influence Functions | $2C + O(N^2)$ (full training + expensive Hessian computations + final training) |
| Data Shapley | 11C–101C (C for final training + 10C–100C for Monte Carlo approximation) |
| Probabilistic Bilevel Optimization | 50C–100C (requires 50–100 outer iterations) |
| Greedy Coreset | $\sim (N^2/T + 1)$C where K is coreset size |

**Summary:** CADS achieves superior efficiency among bilevel methods with overhead $(5/A + 2\gamma)$C. With A=5 amortization, total cost reduces to $2\gamma$C, $\gamma < 1$. Traditional methods like Forgetting Scores offer moderate 1.2C cost but lack budget-aware optimization and deliver inferior effectiveness. Advanced bilevel methods become prohibitively expensive (50C+), while CADS provides an optimal balance between computational efficiency and budget-aware capabilities.

**Note:** The core contribution of our work is introducing computational budget constraints into coreset selection, rather than optimizing selection time alone. Once learned, a coreset can be reused multiple times across different training scenarios, model architectures, and experiments. Therefore, the computational investment in coreset selection can be amortized over multiple uses, making the selection efficiency a secondary consideration compared to the quality of the resulting coreset under budget constraints.

## 6 Conclusion

We introduced CADS, a compute-aware data-selection framework that casts subset choice as bilevel optimization under explicit budget constraints. By combining a probabilistic reparameterization with a Hessian-free policy-gradient estimator and a penalty-based surrogate that reduces inner-loop optimization to a one-dimensional loss, CADS efficiently balances data quantity, quality, and distribution. Our example-level (CADS-E) and source-level (CADS-S) variants deliver up to 14.42% accuracy gains and 3–20× speedups over fixed-budget baselines on vision and language benchmarks. Future work will extend CADS to jointly adapt model size, dataset scale, and compute budget for fully resource-adaptive training.

## Acknowledgments and Disclosure of Funding

This work was supported by High-Quality Development Project of Shanghai Municipal Commission of Economy and Informatization (Grant No. 2024-GZL-RGZN-02010), AI for Science Foundation of Fudan University (FudanX24AI028), and the National Nature Science Foundation of China (62472097).

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

# Supplemental Material: Computational Budget Should Be Considered in Data Selection

In this appendix, we provide comprehensive additional materials to supplement the main text. The contents include:

- **Broader impacts (Section A):** A discussion on the broader implications of our research.

- **Experimental setting and implementation details (Section B):** Detailed information on the experimental settings and implementation details.

- **Log-linear surrogate of compute-constrained training loss (Section C):** Explanation of approximating $\mathcal{L}_{\mathrm{trn}}(\mathbf{u}, \mathbf{m})$ by the scale-dependent surrogate $l(|\mathbf{m}|)$, its log-space interpolation, and fitting procedure.

- **Visualizations on truncated Gaussian distribution (Section D):** Detailed visualization on the truncated Gaussian distribution.

- **Additional experimental results (Section E):** MNIST experiments varying the total compute budget to compare random, PBCS, CADS-E, and Bilevel-CADS. The results show that our method consistently achieves the highest accuracy (Table 9).

- **Time efficiency analysis (Section F):** Benchmarking of average training and sampling runtimes on representative hardware.

- **Scalability to other bilevel optimization problems (Section G):** Discussion on the applicability conditions, loss estimation accuracy, and a validation framework for applying CADS to new domains.

- **Limitations and future work (Section H):** Analysis of the limitations in our current framework, and plans for future improvements.

- **Licenses for existing assets (Section I):** Acknowledgment and respect for the licenses and terms of use of datasets and code libraries utilized in our research.

## A  Broader impacts

This work introduces CADS, a compute-aware data selection algorithm that dynamically adapts the training data budget according to available computational resources. By optimizing data efficiency relative to compute constraints, CADS significantly reduces the computational overhead of training deep learning models without sacrificing performance. This advancement enables more accessible and environmentally sustainable machine learning research and deployment, particularly in scenarios constrained by limited hardware resources. The method's potential to lower energy consumption during model training aligns with broader community efforts towards greener AI, contributing to reductions in carbon footprint and operational costs. Additionally, CADS facilitates democratization of deep learning by empowering researchers and practitioners with modest resources to train competitive models. However, as with any automation that accelerates model training, there exist potential risks, including the possibility of speeding up the development of models with harmful biases, privacy vulnerabilities, or malicious intent if not applied responsibly. We stress the importance of ethical use and compliance with community norms and regulatory standards when deploying such techniques. Overall, CADS represents a step forward in efficient and responsible machine learning practices, promoting sustainability and equitable access within the deep learning community.

## B  Experimental setting and implementation details

In this section, we detail the protocols and implementation specifics underlying all our evaluations. All experiments were run on a single machine equipped with an NVIDIA A100 80 GB GPU under CUDA 12.6 and NVIDIA driver 470.199.02, using PyTorch 2.5.1.

## B.1 Exploratory experiment on spectrum bias

To isolate the effect of spectral content on generalization, we generate two synthetic datasets with identical noise level $\sigma = 0.1$ but different frequency composition.

**Data generation.** We sample $N = 50{,}000$ inputs for each group; for the low-frequency dataset (Group A) we set $y = x + \mathcal{N}(0, \sigma^2)$, and for the high-frequency dataset (Group B) we set $y = x + \frac{\sin(\pi x)}{\pi x} + \mathcal{N}(0, \sigma^2)$. A fixed validation set of 10,000 points is used to track generalization error.

**Model and training.** Each model is a three-layer MLP (100–100 hidden units, ReLU). We train for 160 steps using Adam (learning rate $3 \times 10^{-4}$) and batch size 1000. After every parameter update, we compute the MSE on the validation set and log the result. The different positions on the x-axis in Figure 1 correspond to these intermediate results recorded during the training process.

## B.2 Small-scale validation

We validate CADS on a reduced-scale MNIST classification task using a simple convolutional network. All experiments use a fixed training set of 1,000 examples and batch size 1,000. We optimize under an epoch-based budget of 20 epochs. The selection parameters $s \in \mathbb{R}^{|D|}$ are initialized as $s = v \cdot \mathbf{1}$, $v \in \{0.2, 0.4, 0.6, 0.8\}$, $\mathbf{1} \in \mathbb{R}^{|D|}$ is the all-ones vector. We solve the bilevel problem with Adam for both the network parameters $\theta$ (learning rate $5 \times 10^{-3}$) and the selection weights $s$ (learning rate $5 \times 10^{-2}$). The outer loop runs for 300 iterations with variance reduction and gradient clipping enabled.

**Network architecture.** We employ a simple convolutional model: two convolutional layers with $5 \times 5$ kernels and channel counts $\{32, 64\}$, each followed by ReLU and optional $2 \times 2$ max-pooling; the feature map is flattened ($4 \times 4 \times 64 = 1024$ units) and passed through a fully connected layer of 128 units with ReLU; a final linear layer outputs class logits. Input normalization by fixed mean and standard deviation is applied when enabled. No dropout is used.

## B.3 Heterogeneous data sources: small-scale experiments on CIFAR-10

We evaluate CADS-S on a grouped CIFAR-10 variant with five demographic groups and up to $90\%$ label noise. All experiments use the full training set (no limit) with batch size 256. We allocate an epoch-based compute budget of 2 to 5 epochs. The selection parameter $s$ is initialized to $0.5 \cdot \mathbf{1}^{|D|}$. We solve the bilevel problem with Adam for both network parameters $\theta$ (learning rate $5 \times 10^{-3}$) and selection weights $s$ (learning rate $5 \times 10^{-2}$), over 100 outer iterations with variance reduction and gradient clipping. We adopt the standard ResNet-18 backbone for all runs.

## B.4 Heterogeneous data sources: scaling to larger datasets

We evaluate CADS-S on DomainNet with 6 groups using the full training set and batch size 1024. We allocate an epoch-based compute budget of 10 epochs. The selection parameters $s \in \mathbb{R}^{|D|}$ are initialized as $s = v \cdot \mathbf{1}$, $v \in \{0.2, 0.4, 0.6, 0.8\}$, $\mathbf{1} \in \mathbb{R}^{|D|}$ is the all-ones vector. We solve the bilevel problem with Adam for both network parameters $\theta$ (learning rate $5 \times 10^{-3}$) and selection weights $s$ (learning rate $5 \times 10^{-2}$), over 100 outer iterations with variance reduction and gradient clipping. We adopt the standard ResNet-18 backbone for all runs.

**DomainNet dataset.** DomainNet comprises over 0.5 million images across six visual domains: Real, Sketch, Clipart, Painting, Infograph, and Quickdraw (As shown in Fig. 6). In our setup, we treat the Real domain as a high-quality source but include only 10,000 samples from it to simulate limited access; the other five domains are used in full.

## B.5 Instruction tuning tasks: small-scale experiments

We evaluated our CADS-S approach on instruction fine-tuning tasks using the GPT-2 model [48]. The dataset consists of two heterogeneous data sources: (i) **Alpaca-GPT4** [46], a high-quality dataset from GPT-4-generated instructions, from which we sample 1,000 examples to simulate the rarity

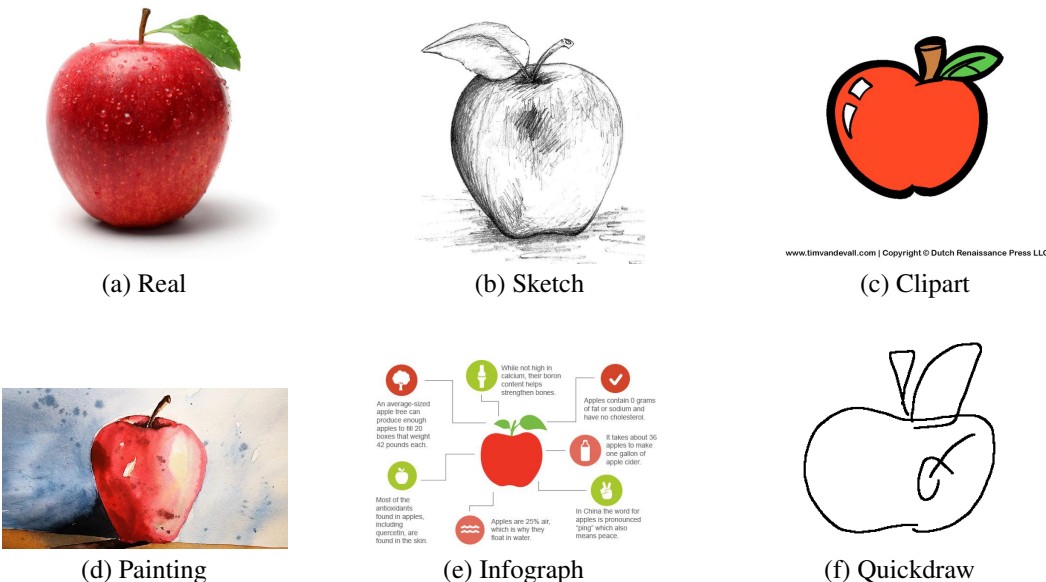

(a) Real        (b) Sketch        (c) Clipart

(d) Painting      (e) Infograph      (f) Quickdraw

Figure 6: Representative examples from the DomainNet domains: Real, Sketch, Clipart (top row); Painting, Infograph, Quickdraw (bottom row).

of premium data sources in practical scenarios; (ii) **Alpaca** [57], a larger dataset containing 9,000 examples, representing standard-quality data. The total compute budget varies within the range $[1 \times 10^4, 5 \times 10^4]$ sample usages. The selection weight $s$ is initialized to $0.5 \cdot \mathbf{1}^{|D|}$. We solve the bilevel problem with AdamW for both model parameters $\theta$ (learning rate $1 \times 10^{-5}$) and selection weights $s$ (learning rate $5 \times 10^{-2}$). We adopt the GPT-2 backbone with maximum sequence length 1024 for all runs.

**Data preprocessing.** Raw instruction–response samples are loaded from JSON or JSONL files and consolidated into a flat record list. A pretrained tokenizer is initialized and extended to include five special markers: a padding token, an end-of-sequence token, and three role-demarcation tokens (<|system|>, <|user|>, <|assistant|>). Each record is then serialized into a single text sequence by:

1. Emitting the system token, followed by the system-level directive, and two line breaks.

2. Emitting the user token, followed by the instruction text (and any optional input), and two line breaks.

3. Emitting the assistant token, followed by the target response, and terminating with the end-of-sequence token.

The concatenated text is first tokenized without truncation to measure its length; any example that exceeds the maximum allowed token count is discarded. The remaining examples are split into training, validation, and test subsets according to either explicitly provided sizes or a default 90/5/5 ratio. At training time, each sequence is tokenized with truncation to the maximum length, producing token identifiers and attention masks. All token positions corresponding to the system and user segments are assigned an ignore index in the label sequence so that only assistant-response tokens contribute to the loss. A bespoke collation routine then pads every batch to the length of its longest sequence, using the padding token for inputs, zeros for masks, and the ignore index for both prompt and padding positions in the labels.

### B.6  Instruction tuning tasks: scaling to additional datasets

We extended our evaluation to 13 distinct instruction-following corpora, drawing ten thousand examples from each. These included the original AlpacaGPT4 benchmark[46], the SlimOrca set[32], the Alpaca collection[57], the GPTeacher suite[58], and nine multilingual variants of the Alpaca data.

In total, approximately 130 000 samples were processed with the identical tokenization, formatting, filtering, and batch-collation pipeline described above. The complete list of datasets, along with their sizes and licenses, is summarized in Table 7.

| Dataset | Size | License |
|---|---|---|
| AlpacaGPT4 [46] | 52K | Apache-2.0 |
| SlimOrca [32] | 518K | MIT |
| GPTeacher [58] | 89K | MIT |
| Alpaca [57] | 52K | Apache-2.0 |
| Alpaca-es[2] | 52K | CC-BY-4.0 |
| Alpaca-de[3] | 50K | Apache-2.0 |
| Alpaca-ja[4] | 52K | CC-BY-NC-SA-4.0 |
| Alpaca-ko[5] | 50K | CC-BY-NC-4.0 |
| Alpaca-ru[6] | 30K | CC-BY-4.0 |
| Alpaca-it[7] | 52K | CC-BY-NC-SA-4.0 |
| Alpaca-fr[8] | 55K | Apache-2.0 |
| Alpaca-zh[9] | 49K | CC-BY-4.0 |
| Alpaca-pt[10] | 52K | CC-BY-NC-4.0 |

Table 7: Summary of high-quality instruction-tuning datasets (top) and multilingual Alpaca variants (bottom).

## C Log-linear surrogate of compute-constrained training loss

### C.1 Data Collection

We collect estimates of the $K$-step ("compute-constrained") training loss $\mathcal{L}_{\mathrm{trn}}(\theta; |m|)$ at a range of subset sizes $|m|$. By default we choose nine relative fractions $\{0.01, 0.02, 0.05, 0.1, 0.3, 0.5, 0.7, 0.9\}$ of the full training set (clamped to at least 50 examples on MNIST or the configured batch-size otherwise). For each size we train for a total compute budget $N$ (so that each inner training run sees roughly the same total number of gradient steps), and record the final training loss. We then summarize each size by its empirical $\ell_j$.

### C.2 Surrogate model choice

We set $\varepsilon = 10^{-10}$ and fit the transformed log-loss $\log(l(|m|) + \varepsilon)$ using two options:

- **Linear:** $\log(l(|m|) + \varepsilon) = k\,|m| + b$.

- **Cubic spline interpolation:** fit a cubic spline through the points $(|m_j|, \log(l(|m_j|) + \varepsilon))$, yielding a piecewise-cubic function $\hat{f}(|m|)$.

---

[2]https://huggingface.co/datasets/bertin-project/alpaca-spanish
[3]https://huggingface.co/datasets/mayflowergmbh/alpaca-gpt4_de
[4]https://huggingface.co/datasets/fujiki/japanese_alpaca_data
[5]https://huggingface.co/datasets/Bingsu/ko_alpaca_data
[6]https://huggingface.co/datasets/IlyaGusev/ru_turbo_alpaca
[7]https://huggingface.co/datasets/mchl-labs/stambecco_data_it
[8]https://huggingface.co/datasets/jpacifico/French-Alpaca-dataset-Instruct-55K
[9]https://huggingface.co/datasets/llm-wizard/alpaca-gpt4-data-zh
[10]https://huggingface.co/datasets/dominguesm/alpaca-data-pt-br

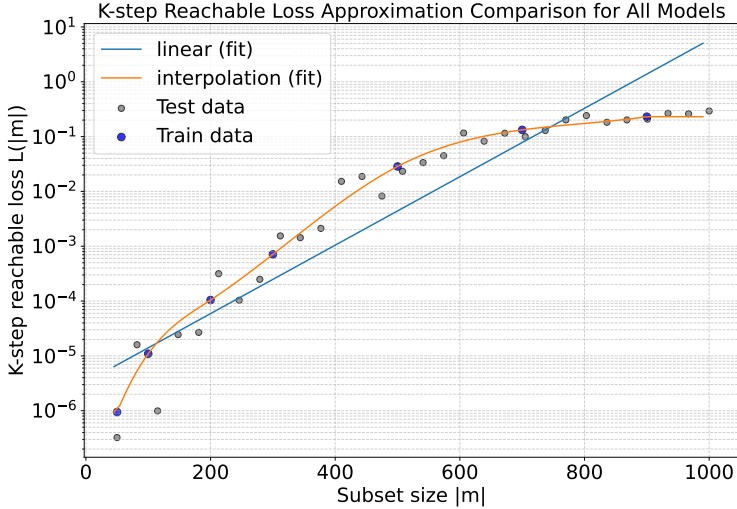

Figure 7: All-models comparison of the K-step reachable loss as a function of subset size $|m|$. Blue circles denote observed mean loss on training subsets; gray squares denote observations on held-out test subsets. Solid curves show the fitted approximation functions for each candidate model. The vertical axis is plotted on a logarithmic scale.

After fitting in log-space, we recover the original-scale surrogate via $l(|m|) = \exp\big(\hat{f}(|m|)\big)$. Figure 7 overlays these fitted curves on the empirical training-loss measurements. The cubic spline interpolation achieves a noticeably lower MSE, motivating its use when maximum fidelity is desired, while the linear surrogate offers simplicity and a closed-form gradient. **Since CADS never computes $\partial l(|m|)/\partial|m|$, we exclusively employ the cubic spline interpolation due to its superior fit.**

### C.3  Analysis of Sampling Efficiency for Loss Estimation

To evaluate the sample efficiency of our loss estimation method, we conducted an analysis to determine the minimum number of subset samples ($K$) required for reliable interpolation.

**Experimental Design**  On the CIFAR-10 dataset with a ResNet-18 model, we evaluated the loss estimator's performance under varying sampling densities, where $K \in \{4, 5, 6, 7, 8\}$. For each value of $K$, we sampled $K$ distinct subset sizes to train corresponding models, yielding $K$ pairs of (size, loss) data points. These points were used to fit the loss estimator. Subsequently, we tested each fitted estimator on a held-out set of 100 separately sampled (size, loss) pairs to measure its interpolation accuracy.

**Results**  The results, summarized in Table 8, demonstrate that the estimator's performance improves significantly up to $K = 5$ sampling points, after which the returns diminish. This indicates that our method is highly sample-efficient. Notably, increasing $K$ further (e.g., $K = 7$) can slightly degrade performance, likely due to overfitting the estimator to a specific set of sample points, which hinders its ability to generalize to unseen subset sizes.

## D  Visualizations on truncated Gaussian distribution

In this section we provide visual intuition for the source-level sampling prior by plotting the truncated Gaussian density $p(r \mid s)$ on the interval $[0, 1]$ for a variety of center values $s$ and scales $\sigma$. Figures 8, 9, and 10 illustrate how annealing $\sigma$ concentrates the distribution around its mean.

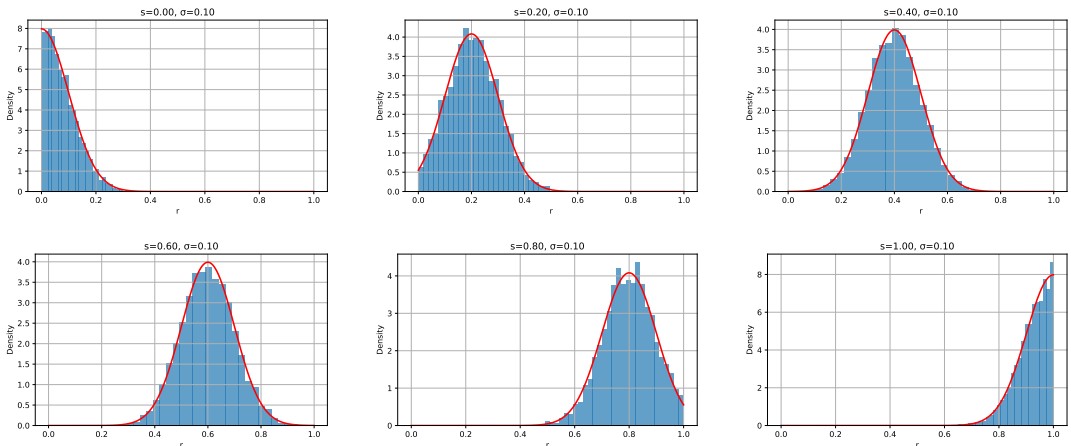

Figure 8: Sampling prior $p(r \mid s)$ on $[0, 1]$ for various center values $s$ with scale $\sigma = 0.1$.

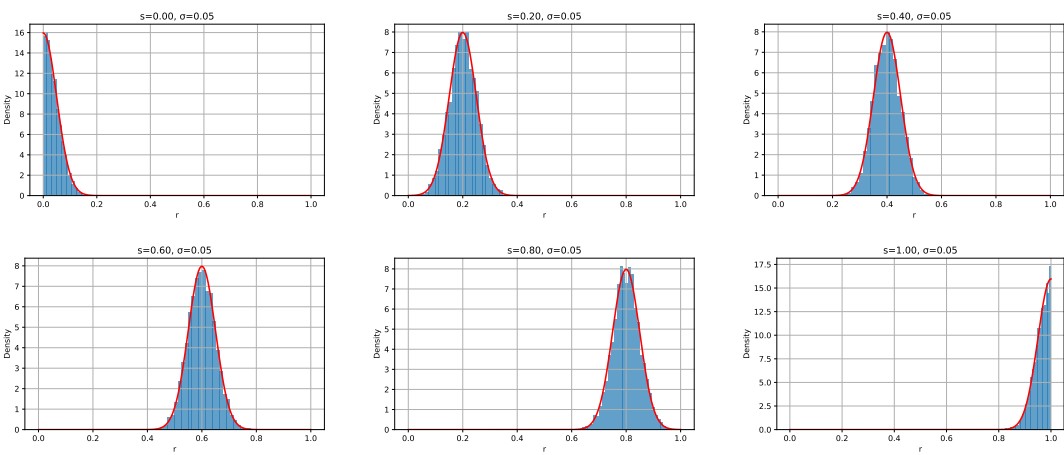

Figure 9: Sampling prior $p(r \mid s)$ on $[0, 1]$ for various center values $s$ with scale $\sigma = 0.05$.

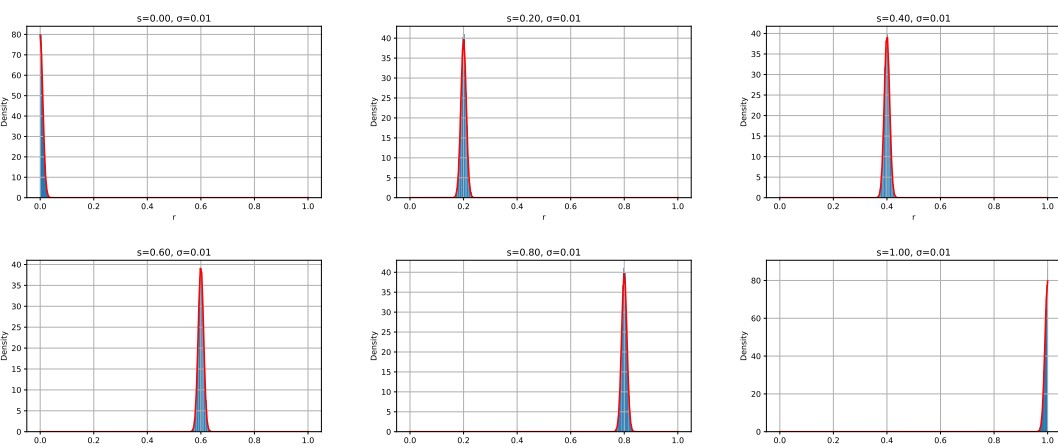

Figure 10: Sampling prior $p(r \mid s)$ on $[0, 1]$ for various center values $s$ with scale $\sigma = 0.01$.

Table 8: Mean Square Error (MSE) of the loss estimator with respect to the number of sampling points ($K$). Performance saturates at $K = 5$, demonstrating high sample efficiency.

| Sampling Points ($K$) | Mean Square Error |
|:---:|:---:|
| 4 | 0.057668 |
| 5 | 0.016326 |
| 6 | 0.018594 |
| 7 | 0.020031 |
| 8 | 0.017905 |

## E  Additional experimental results

**Additional result on MNIST**    The purpose of this experiment is to investigate the effect of compute budget on data selection within the MNIST dataset. We vary the compute budget in terms of total forward counts (10 000, 20 000, 50 000 and 100 000). At each budget we compare four sampling strategies—random, PBCS, CADS-E and our Bilevel-CADS. As shown in Table 9, our Bilevel-CADS consistently outperforms all baselines in all compute budgets.

Table 9: Results on MNIST with various compute budgets.

| Method | Compute budget | | | | Average |
|---|---|---|---|---|---|
| | 10,000 | 20,000 | 50,000 | 100,000 | |
| Random | 87.36 | 88.05 | 87.71 | 87.24 | 87.59 |
| PBCS | 89.61 | 90.48 | 90.96 | 90.41 | 90.37 |
| CADS-E | 90.62 | 91.48 | 92.45 | 93.17 | 91.93 |
| Bilevel-CADS | **90.70** | **92.44** | **93.09** | **93.61** | **92.46** |

## F  Time efficiency analysis

### F.1  Computational complexity compare to standard bilevel optimization

We compare the per-iteration cost of our single-epoch inner loop against the full-training inner loop used in bilevel-CADS, and then account for the one-time cost of fitting the loss estimator $l(|\boldsymbol{m}|)$.

Let

- $|\mathcal{D}|$ be the size of the training set,
- $T_{\text{ep}} = O(|\mathcal{D}|)$ the cost of one full epoch (one forward+backward pass),
- $N$ the number of epochs used by bilevel-CADS in its inner optimizer,
- $K$ the number of subsets sampled per outer iteration,
- $|\boldsymbol{m}|$ the average subset size,
- $M$ the total number of outer iterations in a run.

**Bilevel-CADS inner solve (per outer step)**

$$\text{Cost}_{\text{bilevel}} \ = \ K \times N \times T_{\text{ep}} \ = \ O\big(KN|\mathcal{D}|\big).$$

**Our CADS-E/S inner loop (per outer step, ignoring estimator)**

$$\text{Cost}_{\text{CADS}}^{\text{base}} \ = \ K \times T_{\text{ep}} \ = \ O\big(K|\mathcal{D}|\big).$$

**Estimator fitting overhead**    To fit the mapping $l(|\boldsymbol{m}|)$ we train 8 models for $N$ epochs each (As detailed in  D), incurring a one-time cost

$$\text{Cost}_{\text{est}} \ = \ 8 \times N \times T_{\text{ep}}.$$

Amortized over $M$ outer steps, this adds

$$\frac{\text{Cost}_{\text{est}}}{M} = \frac{8\,N\,T_{\text{ep}}}{M}$$

to each iteration.

**Total CADS cost (amortized)**

$$\text{Cost}_{\text{CADS}} = K \times T_{\text{ep}} + \frac{8\,N\,T_{\text{ep}}}{M} = T_{\text{ep}}\Big(K + \tfrac{8\,N}{M}\Big).$$

**Speed-up factor**

$$\frac{\text{Cost}_{\text{bilevel}}}{\text{Cost}_{\text{CADS}}} = \frac{K\,N\,T_{\text{ep}}}{T_{\text{ep}}\big(K + \tfrac{8\,N}{M}\big)} = \frac{K\,N}{K + 8\,N/M}.$$

In our experiments $K = 5$ and $M = 100$, giving

$$\frac{\text{Cost}_{\text{bilevel}}}{\text{Cost}_{\text{CADS}}} = \frac{1}{1/N + 0.016}.$$

In practice, larger $N$ yields an even greater speed-up.

## F.2 Empirical Validation

We measure end-to-end wall-clock time of bilevel-CADS and CADS-E on MNIST as a function of compute budget $C$ (total epochs). We sweep $C$ over $\{5, 50, 100, 200, 500, 1000, 2000, 5000\}$ and record the runtime of each method under identical hardware. Figure 11 confirms a roughly linear scaling for both methods, with CADS-E exhibiting a much smaller slope.

## F.3 Analysis of Selection Algorithm Runtime

A potential concern regarding our method is that the runtime of the selection algorithm may scale with the number of sampled subsets ($K$), which could offset the computational efficiency gains. We address this concern by highlighting several key aspects of our approach.

**Minimal Sampling for Practical Implementation**  Our policy gradient approach for coreset selection requires only a small number of samples ($K$) to ensure training stability. For instance, the comparable PBCS method [74] utilizes $K = 1$ for all its experiments. Our results show that using $K = 2$ is sufficient to achieve strong performance while maintaining high computational efficiency. In cases where smaller $K$ values might lead to higher gradient variance (e.g., in more complex problems), this can be mitigated using established variance reduction techniques. A common approach is to use a self-critical baseline, which adjusts the reward signal as follows:

$$\tilde{\mathcal{L}}(S_i) = \mathcal{L}(S_i) - \frac{1}{K}\sum_{j=1}^{K}\mathcal{L}(S_j). \tag{12}$$

**Amortized Cost of Coreset Selection**  It is important to emphasize that the core contribution of this work is the introduction of computational budget constraints into the coreset selection process, rather than optimizing the selection runtime itself. Once generated, a coreset is a reusable asset that can be applied across various training scenarios, different model architectures, and subsequent experiments. Consequently, the initial computational investment in the selection process is amortized over these multiple uses. This makes the selection efficiency a secondary consideration compared to the quality and utility of the resulting budget-constrained coreset.

## G   Scalability to Other Bilevel Optimization Problems

CADS is designed to address a fundamental challenge in bilevel optimization: scenarios where the inner-level problem's convergence is hindered by computational budget constraints. Many practical bilevel problems suffer from this issue, where traditional methods that assume unlimited computational resources for inner-level convergence often fall short. We have also successfully applied a CADS-like method to resource-intensive diffusion models, demonstrating its potential beyond the scope of this work. The scalability of CADS to other bilevel problems primarily depends on the following factors:

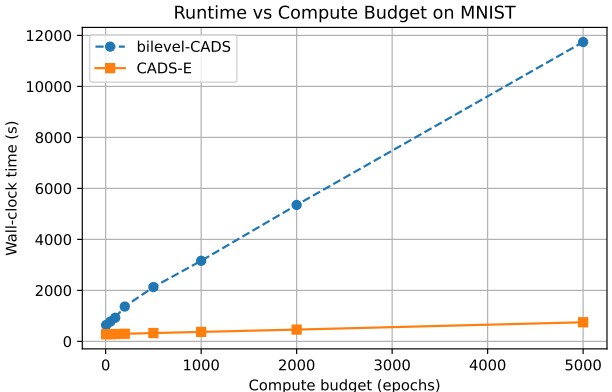

Figure 11: Wall-clock runtime vs. compute budget $C$ for bilevel-CADS (dashed) and CADS-E (solid), MNIST.

**Applicability Conditions**   The core premise of CADS is most relevant when the inner-level optimization is computationally expensive and cannot be fully converged. Our approach provides a framework to handle such resource-constrained scenarios, which are common in real-world applications but often overlooked by conventional methods.

**Loss Estimation Accuracy**   The effectiveness of CADS in a new domain hinges on the accuracy of the loss estimator. As demonstrated in our analysis of loss estimation generalization, the proposed estimator maintains relatively stable predictive performance across the diverse experimental domains explored in this paper. This stability suggests that similar performance may be achievable in other domains with comparable data characteristics.

**Validation Framework for New Domains**   To adapt CADS to a new bilevel optimization problem, we propose the following validation framework:

- **Loss Estimator Generalization Assessment.** Before full-scale implementation, it is crucial to assess whether the data distribution characteristics of the new domain can be reliably predicted. This can be achieved by conducting experiments similar to our generalization evaluation to validate the feasibility of loss estimation.
- **Domain-Adaptive Estimator Design.** While our current estimator is effective for data selection problems, different bilevel scenarios might benefit from specialized estimators. The architecture of the loss estimator should be tailored to the specific requirements and data characteristics of the target domain to ensure optimal performance.

## H   Limitations and Future Work

Similar to existing methods, our approach is based on standard training methodologies, but in the industry, large language models incorporate numerous engineering optimizations. Consequently, our budget is likely proportional to theirs, rather than perfectly aligned. For instance, in sparse mixture of experts (MoE), the budget is determined by model capacity and routing complexity, whereas we measure performance by the number of forward passes. This measurement may not fully correspond with the budget metrics used in industry. However, we firmly believe that our methods are applicable in industrial contexts; we simply need to adjust the budget measurements accordingly.

## I   Licenses for existing assets

We rely on several public datasets and open-source libraries, and all original authors are properly credited and their licenses fully respected. The vision benchmarks we employ are MNIST (public domain), CIFAR-10 (MIT License) and DomainNet (CC BY-NC-SA 4.0). Our instruction-tuning

corpora are summarized in Table 7, with licenses ranging from Apache 2.0 to various Creative Commons and MIT terms. On the software side, we build atop PyTorch and torchvision (BSD 3-Clause), SciPy (BSD), and HuggingFace Transformers (GPT-2, Apache 2.0); ResNet models are taken from torchvision (BSD 3-Clause). We confirm that every dataset and code dependency is used in accordance with its license and citation requirements.

