# OpenReview forum: "Computational Budget Should Be Considered in Data Selection"
_NeurIPS.cc/2025/Conference — NeurIPS 2025 poster_

### Official Review · Reviewer_cbM7 · 2025-06-29

**Clarity:** 2
**Significance:** 3
**Originality:** 3
**Rating:** 4
**Confidence:** 5

**Summary:**

The paper introduces a framework that incorporates **computational budget constraints** into data selection, presenting the Computational Budget-Aware Data Selection (CADS) method. Formulated this as a bilevel optimization problem, CADS consists of an inner loop that trains a model on a selected data subset within a defined computational budget and an outer loop that optimizes data selection based on the model's performance.

The authors address two key challenges in solving this bilevel problem: **the high computational cost of Hessian estimation and the complexity of achieving inner-loop optimality**. To overcome these, CADS employs a probabilistic reparameterization strategy with a Hessian-free policy gradient estimator and reformulates the inner optimization as a penalty term in the outer objective, requiring only a one-dimensional loss estimation for enhanced efficiency. This loss estimation is performed using a log-linear function interpolated from losses across different subset sizes.

CADS is implemented in two variants: CADS-E, which enables fine-grained, example-level selection, and CADS-S, which offers group-level sampling ratio estimation. Experimental results across vision (MNIST, CIFAR10, DomainNet) and language tasks (instruction-tuning dataset) demonstrate that CADS achieves up to 14.42% performance improvements and 3–20x training speedups compared to baseline methods, showcasing its effectiveness and efficiency.

**Questions:**

* The definitions of low-frequency and high-frequency data are unclear, particularly how these characteristics are identified or quantified in the context of the dataset (image or instruction-tuning dataset).
* In Section 3, how do we construct validation dataset? how it is sourced from Group A and Group B, as referenced in Appendix B.1?
* What is the dataset and model being used to have Figure 7?

**Ethical Concerns:**

["NO or VERY MINOR ethics concerns only"]

**Final Justification:**

Reviewer's response during rebuttal resolved my concern to this paper.

**Limitations:**

Yes. It's documented in their appendix (see section A and G).

**Paper Formatting Concerns:**

It's following correct format.

**Quality:**

2

**Strengths And Weaknesses:**

[Strengths]
* Quality: The first half of the paper (up to Section 5) is well-structured, effectively motivating the reader with a clear rationale for the proposed approach. However, the experimental section is less clear, making it difficult to follow the setup and key takeaways from each experiment. I notice that Section 5.2, which discusses the CIFAR-10 experiment, missing results referencing the relevant table, only the experimental setup.
* Clarity: The mathematical formulation is clear, making the challenges easy to understand.

[Weaknesses]
* About runtime concerns, while incorporating computational budget constraints is a strength, the runtime of the selection algorithm may scale poorly with the number of subsets (K), potentially offsetting the computational efficiency gains of the method. For example, in algorithm 1, we have to sample K subsets and train K models and evaluate them to access K losses so we can interpolate.
* Figure 7, for example in the appendix illustrates the loss estimation process. It remains unclear about sampling requirements, especially how many subsets (K) are needed to generate sufficient sample losses for reliable interpolation and how robust this interpolation.
* It's unclear of the selection of the baseline method (PBCS) or provide details on how it operates. Could the authors clarify it?

---

> ### Author Rebuttal · Authors · 2025-07-30
>
> **We sincerely thank Reviewer cbM7 for the constructive feedback.** Below we provide point-by-point responses:
>
> ---
>
> ### **S1: I notice that Section 5.2, which discusses the CIFAR-10 experiment, missing results referencing the relevant table, only the experimental setup.**
>
> Thank you for pointing this out. We will fix this in the revision.
>
> ---
>
> ### **W1: About runtime concerns, while incorporating computational budget constraints is a strength, the runtime of the selection algorithm may scale poorly with the number of subsets (K), potentially offsetting the computational efficiency gains of the method.**
>
> Thank you for this important concern. We'd like to clarify several key aspects of our approach that address this issue:
>
> ### **[K=2 is Sufficient for Practical Implementation]**
>
> 1. **Minimal K requirement**: For the policy gradient in coreset selection, a small K is always sufficient to ensure training stability. For exmaple, PBCS [74] adopts K=1 for all its experiments.  Our results in table below show that **K=2 is sufficient to achieve strong performance while maintaining computational efficiency.**
> 2. **Variance reduction technique [r1]**: When dealing with more complex problems, smaller values of K may lead to higher gradient variance, and in such cases, we can leverage well-established variance reduction techniques to effectively mitigate this issue:
>       $$\tilde{L}(S_i) = L(S_i) - \frac{1}{K}\sum_{j=1}^K L(S_j).$$
>
>
> **Note:** The core contribution of our work is introducing computational budget constraints into coreset selection, rather than optimizing selection time alone. Once learned, **a coreset can be reused multiple times** across different training scenarios, model architectures, and experiments. Therefore, the computational investment in coreset selection can be amortized over multiple uses, making the selection efficiency a secondary consideration compared to the quality of the resulting coreset under budget constraints.
>
> [r1] Hammersley, J. M.; Handscomb, D. C. (1964). Monte Carlo Methods. London: Methuen. ISBN 0-416-52340-4.
>
> ### **[Empirical Validation]**
>
> We provide empirical evidence that K=2 achieves comparable performance to larger K values:
>
> | Initial subset size (same as Tab.1 in main text) | K=2        | K=5        |
> | ------------------------------------------------ | ---------- | ---------- |
> | 200     | 91.44%     | 91.48%     |
> | 400      | 92.23%     | 92.28%     |
> | 600       | 92.38%     | 92.34%     |
> | 800          | 92.88%     | 92.90%     |
> | **Average**              | **92.23%** | **92.25%** |
>
> **Note:** We provide an analysis of the computational overhead of CADS in our reply to reviewer **TtXp**.
>
> ---
>
> ### **W2: For Figure 7, how many subsets (K) are needed to generate sufficient sample losses for reliable interpolation and how robust this interpolation.**
>
> ### **[Sampling Efficiency Analysis]**
>
> We conducted experiments to demonstrate that our method requires only a **small number of sampling points** for reliable loss estimation.
>
> **Experiment Design:** On ResNet-18 and CIFAR-10, we evaluate loss estimators under different sampling densities K ∈ {4, 5, 6, 7, 8}. For each K, we sample K subset sizes to train models and collect K (size, loss) pairs for fitting the loss estimator. We then test each estimator on 100 separately sampled (size, loss) pairs to evaluate interpolation accuracy across different sampling densities.
>
> **Results:**
>
> | Sampling Points (K) | Mean Square Error |
> | ------------------- | ----------------- |
> | 4        | 0.057668          |
> | **5**        | **0.016326**      |
> | 6         | 0.018594          |
> | 7        | 0.020031          |
> | 8         | 0.017905          |
>
> The results show that **performance saturates around K=5 sampling points**, with diminishing returns beyond this point. This demonstrates our method's sample efficiency.
>
> Notably, **larger K values can actually hurt performance** (as seen with K=7) due to overfitting to the specific sampled points, reducing generalization to unseen subset sizes.
>
> ### **[Practical Implementation]**
>
> In practice, we recommend:
>
> - **K=5 sampling points** for most applications (optimal efficiency)
> - **Logarithmic spacing** of subset sizes for better coverage
>
> ---
>
> ### **W3: It's unclear of the selection of the baseline method (PBCS) or provide details on how it operates. Could the authors clarify it?**
>
> We appreciate the reviewer's request for clarification on the PBCS baseline method.
>
> ### **[Why We Choose PBCS as Baseline]**
>
> PBCS is selected as our primary baseline because it represents the **most representative and state-of-the-art bilevel optimization approach** for data selection problems.
>
> ### **[How it operates]**
>
> PBCS proposes a **probabilistic bilevel coreset selection framework** that differs fundamentally from our computational budget-aware approach. PBCS formulates coreset selection as a **continuous bilevel optimization problem** using probabilistic reparameterization:
>
> **Key Formulation:**
>
> $$\min_{s\in C} \Phi(s) = \mathbb{E}_{p(m|s)} \mathcal{L}(\theta^*(m))$$
>
> $$\text{s.t. } \theta^*(m) \in \arg \min_\theta \hat{\mathcal{L}}(\theta; m)$$
>
> Where:
>
> - Each training sample $i$ is assigned a **Bernoulli probability $s_i$** to be included in the coreset
> - Binary mask $m_i \sim \text{Bern}(s_i)$ determines actual sample selection
> - Constraint set $C = \{s : 0 \preceq s \preceq 1, \|s\|_1 \leq K\}$ controls coreset size $K$
>
> **Optimization Strategy:**
> PBCS uses a PGE to avoid expensive implicit differentiation:
> $$\nabla_s \Phi(s) \approx \mathcal{L}(\theta^*(m)) \nabla_s \ln p(m|s)$$
>
> The algorithm alternates between:
>
> 1. **Inner loop**: Sample coreset from $p(m|s)$ and train model to convergence
> 2. **Outer loop**: Update probabilities $s$ using PGE based on full dataset performance
>
> ### **[Key Differences Between CADS and PBCS]**
>
> Our CADS framework addresses several fundamental limitations of PBCS:
>
> 1. **Computational Budget Awareness**: Unlike PBCS which treats coreset size as a fixed constraint, our CADS explicitly incorporates **computational budget $C$** as a first-class design consideration, jointly optimizing data selection and budget allocation under resource constraints.
>
> 2. **Inner-loop Non-convergence Handling**: PBCS assumes the inner optimization reaches convergence ($\theta^*(m)$), which is unrealistic under limited computational budgets. Our CADS introduces **penalty-based single-level relaxtion** and **an loss estimators** to handle scenarios where inner-loop training is incomplete.
>
> 3. **Computational Efficiency**: While PBCS requires expensive bilevel optimization with full inner-loop convergence, our CADS **unfolds the bilevel structure** using penalty-based reformulation, achieving **3-20× speedup** by avoiding repeated expensive implicit inner-loop training.
>
> These technical innovations enable CADS to achieve superior performance-efficiency trade-offs compared to traditional size-constrained approaches like PBCS.
>
> ---
>
> ### **Q1: The definitions of low-frequency and high-frequency data are unclear, particularly how these characteristics are identified or quantified in the context of the dataset (image or instruction-tuning dataset).**
>
> Thank you for this feedback. We'd like to clarify the concepts and identification methods below.
>
> ### **[Frequency Definition]**
>
> We follow the concept of **low-frequency** and **high-frequency** given in the literature of deep learning theory [r1]. They are **relative concepts** rather than strict mathematical definitions. This exploratory experiment was designed as a **proof-of-concept** to demonstrate that data preferences can shift with computational budget, which motivated our main contribution.
>
> - **High-frequency data**: More complex, requires more computational resources to effectively utilize.
> - **Low-frequency data**: Simpler patterns that can be learned with less computation.
>
> ### **[How to Identify in Different Dataset Types]**
>
> **Image datasets**: Identified through frequency domain analysis—images with rich edge details and textures contain more high-frequency components, as these represent rapid spatial variations (e.g., edges, fine patterns) that are harder for models to capture [r2]. **This can be quantified using gradient magnitude analysis (e.g., Sobel operators) or Laplacian variance to capture edge density and texture complexity.**
>
> **Instruction-tuning datasets**: High-frequency includes complex multi-step reasoning tasks and domain-specific instructions; low-frequency represents simpler, common instruction patterns.
>
> [r1] N. Rahaman et al. On the spectral bias of neural networks. ICML, 2018.
>
> [r2] Wang et al., "High-Frequency Component Helps Explain the Generalization of Convolutional Neural Networks," CVPR 2020
>
> ---
>
> ### **Q2: In Section 3, how do we construct validation dataset? how it is sourced from Group A and Group B, as referenced in Appendix B.1?**
>
> As detailed in Appendix B.1:
>
> **Validation Dataset Construction:**
> The validation set follows the **same generation process as Group B**: $y = x + sin(\pi x)/\pi x + N(0, \sigma^2)$, with a fixed set of 10,000 points.
>
> **Rationale:**
>
> - **Group A (low-frequency)**: Contains only low-frequency components (y = x + noise)
> - **Group B (high-frequency)**: Contains both low and high-frequency components, where all points from Group A could potentially appear in Group B
> - **Validation set**: Uses Group B's generation process to capture the complete frequency spectrum, providing a comprehensive evaluation benchmark
>
> This design ensures that the validation set represents the full complexity of the target function.
>
> ---
>
> ### **Q3: What is the dataset and model being used to have Figure 7?**
>
> Fig. 7 uses MNIST dataset, and the model is a lightweight CNN network. Specific architectural details are provided in Appendix B.2. **Note** that the results on CIFAR experiments in our response to **W2** also show the effectiveness of our method.

---

> > ### Comment · Reviewer_cbM7 · 2025-08-05
> >
> > I appreciate author's time and response. Thank you for your clarification. I have raised my score.

---

> > > ### Author Response · Authors · 2025-08-05
> > >
> > > Dear Reviewer cbM7,
> > >
> > > Thank you for your appreciation and for raising your score. We are glad our clarifications addressed your concerns, and we will ensure to include them in the final version.

---

### Official Review · Reviewer_vqnc · 2025-07-03

**Clarity:** 3
**Significance:** 3
**Originality:** 3
**Rating:** 4
**Confidence:** 4

**Summary:**

The paper proposes to incorporate a computational budget into the data selection process by proposing a bilevel formulation. The formulation uses a computational constraint to train a model on the subset of the dataset in the lower level, with the upper level optimizing the selection of the subset. Apart from providing insights into how the computational budget affects the selection of data, the paper also presents technical contributions for solving the bilevel problem better than previous methods. Empirical evaluation on datasets like MNISt, DomainNet and instruction tuning tasks shows the effectiveness of the approach.

**Questions:**

See the weaknesses.

**Ethical Concerns:**

["NO or VERY MINOR ethics concerns only"]

**Final Justification:**

The rebuttal and discussions have helped further clarify the importance of the paper's contributions. Thus, I maintain my rating.

**Limitations:**

Yes.

**Quality:**

3

**Strengths And Weaknesses:**

Strengths:
1. The paper addresses a crucial problem of compute-constrained data selection.
2. While the bilevel formulation of the problem is well known the use of the probabilistic reparametrization strategy along with the insight of avoiding the training of the lower level problem to convergence can be useful for solving other bilevel problems.
3. The paper clearly discusses the proposed enhancements to solving the bilevel problems along with the computational complexity analysis of the problem.

Weaknesses
1. The empirical evaluation section of the paper is rather weak with evaluation primarily shown on datasets like Mnist and Domainnet. This makes the paper weak since data selection is desirable on larger and more complicated datasets. It is unclear whether the results presented in the paper inform us about the success of the method on larger datasets.
2. The computational complexity of the method is compared against other bilevel based methods for dataset selection. Adding comparison to other dataset selection methods such as Forgetting scores could be useful.
3. Performance comparison is very limited. Other bilevel approaches as well as non-bilevel approaches should be included in the paper
4. While the related work mentions the paper [67] a direct comparison with their approach is essential.

---

> ### Author Rebuttal · Authors · 2025-07-30
>
> **We sincerely thank Reviewer vqnc for the constructive feedback.** Below we provide point-by-point responses:
>
> ---
>
> ### **W1: Comparison between the proposed CADS and established baselines regarding performance and computational cost on larger dataset.**
>
> We conducted comprehensive ImageNet experiments to address this concern. **Additionally**, in our main paper, we performed experiments on DomainNet, one of the largest multi-domain datasets.
>
> ### **[CADS-E Experiments on ImageNet]**
>
> Due to computational constraints during the rebuttal period, we conducted experiments on ImageNet-100  with 10K samples using ResNet-50. We compared against established baselines including Forgetting, EL2N, GraNd-Score, and Moderate across different computational budgets.
>
> | Method      | Budget: 50K | Budget: 100K | Budget: 200K | Budget: 400K | Final Subset Size         |
> | ----------- | ----------- | ------------ | ------------ | ------------ | ------------------------- |
> | Forgetting  | 81.33%      | 82.39%       | 84.71%       | 86.77%       | [5000,5000,5000,5000]     |
> | EL2N        | 79.18%      | 80.79%       | 83.28%       | 84.82%       | -                         |
> | GraNd-Score | 78.35%      | 79.44%       | 82.10%       | 84.08%       | -                         |
> | Moderate    | 80.61%      | 81.70%       | 84.59%       | 86.84%       | -                         |
> | **CADS-E**  | **82.81%**  | **83.78%**   | **86.46%**   | **88.13%**   | **[4088,6320,7171,8653]** |
>
> CADS-E consistently outperforms all baselines across different computational budgets.
>
> ### **[CADS-S Experiments on ImageNet]**
>
> For CADS-S evaluation, ImageNet-1K partitioned into 5 groups with different label noise levels (0%, 2.5%, 5%, 7.5%, 10%). Experiments were conducted with computational budgets measured in epochs (full dataset traversals). For baseline methods, we combined all 5 groups into a single dataset and applied their selection algorithms to choose 20% of the total data as the training subset.
>
> | Method                     | Budget: 10 epochs              | Budget: 20 epochs              | Budget: 30 epochs              | Budget: 40 epochs              |
> | -------------------------- | ------------------------------ | ------------------------------ | ------------------------------ | ------------------------------ |
> | Forgetting                 | 22.18%                         | 32.87%                         | 39.22%                         | 42.36%                         |
> | EL2N                       | 20.82%                         | 31.35%                         | 38.95%                         | 41.61%                         |
> | GraNd-Score                | 22.47%                         | 32.61%                         | 39.05%                         | 42.79%                         |
> | Moderate                   | 26.31%                         | 35.06%                         | 44.54%                         | 49.89%                         |
> | **CADS-S**                 | **31.20%**                     | **42.14%**                     | **50.58%**                     | **56.13%**                     |
> | **CADS Sampling Ratios r** | [0.85, 0.12, 0.03, 0.00, 0.00] | [0.88, 0.10, 0.02, 0.00, 0.00] | [0.92, 0.06, 0.06, 0.01, 0.00] | [0.95, 0.12, 0.02, 0.02, 0.00] |
>
> ---
>
> ### **W2: Additional comparison to other dataset selection methods.**
>
> We provide comprehensive computational overhead comparison across different dataset selection methods.
>
> ### **[Computational Overhead Framework and Notation]**
>
> We establish unified framework with key parameters:
>
> - **C**: Total compute budget (training cost on full dataset)
> - **N**: Total dataset samples
> - **T**: Training epochs, where T = C/N
>
> All computational overheads are measured relative to the baseline cost **C** of training on the full dataset.
>
> **Note**: For methods requiring pre-trained models, including Forgetting Scores, GraNd, EL2N, we assume the computational cost of obtaining the necessary pre-trained model is **0.2C** (20% of the full training budget), which represents a reasonable approximation for early-stage model training needed to compute gradients or other sample-wise statistics.
>
> ### **[Comparative Analysis of Dataset Selection Methods]**
>
> | **Method**                             | **Total Computational Overhead**                             |
> | -------------------------------------- | ------------------------------------------------------------ |
> | **Random Selection**                   | **C** (no additional cost)                                   |
> | **Forgetting Scores**                  | **1.2C** (0.2C for pre-training and forgetting computation + C for final training) |
> | **GraNd**                              | **1.2C** (0.2C for pre-training and gradient computation + C for final training) |
> | **EL2N**                               | **1.2C** (0.2C for pre-training and error computation + C for final training) |
> | **CADS (Ours)**                        | **(5/A + 2$\gamma$)C** where A is amortization factor                |
> | **Influence Functions**                | **2C + O(N²)** (full training + expensive Hessian computations + final training) |
> | **Data Shapley**                       | **11C~101C** (C for final training + 10C ~ 100C for Monte Carlo approximation with 100~1000 samples) |
> | **Probabilistic Bilevel Optimization** | **50C~100C** (requires 50~100 outer iterations, each involving inner loop model training with computational cost C) |
> | **Greedy Coreset**                     | **~($N^2$/T+1)C** where K is coreset size (bilevel solving for each added sample) |
>
> **Summary:** CADS achieves superior efficiency among bilevel methods with overhead **(5/A + 2$\gamma$)C**. With A=5 amortization, total cost reduces to **2$\gamma$C**, $\gamma < 1$.
>
> Traditional methods like Forgetting Scores offer moderate 1.2C cost but lack budget-aware optimization and deliver inferior effectiveness (e.g., lower accuracy in **W1**).
>
> Advanced bilevel methods become prohibitively expensive **(50C+)**, while CADS provides optimal balance between computational efficiency and budget-aware capabilities.
>
> **Note:** The core contribution of our work is introducing computational budget constraints into coreset selection, rather than optimizing selection time alone. Once learned, **a coreset can be reused multiple times** across different training scenarios, model architectures, and experiments. Therefore, the computational investment in coreset selection can be amortized over multiple uses, making the selection efficiency a secondary consideration compared to the quality of the resulting coreset under budget constraints.
>
> ---
>
> ### **W3: Other bilevel approaches as well as non-bilevel approaches should be included in the paper**
>
> Thank you for this valuable feedback. Please refer to our response to **W1**, where we have provided comprehensive comparisons with numerous methods on ImageNet, demonstrating the effectiveness and efficiency of our approach across different scales and settings.
>
> Regarding other bilevel approaches: Most existing bilevel methods suffer from extremely high computational complexity with expensive second-order computations, making large-scale experiments infeasible within the rebuttal timeframe. Our paper already includes comparison with PBCS, a representative state-of-the-art bilevel method in data selection. Our CADS framework specifically addresses the computational intractability of existing bilevel approaches while maintaining their theoretical advantages through efficient approximation strategies, which aligns with our core contribution of making data selection practical under realistic computational budgets.
>
> ---
>
> ### **W4: Direct comparison with [67].**
>
> Thank you for this question. We want to clarify that **[67] is an empirical evaluation study rather than a methodological contribution**. Their work evaluates various existing data selection methods (BM25, embedding-based, etc.) under different computational budgets and concludes that method performance varies with budget constraints. **They do not propose new algorithms or optimization frameworks**.
>
> ### **[Differences in Research Target]**
>
> **Coverage of [67]:**
>
> - Their study specifically targets LLM instruction tuning with only **single-epoch training** scenarios
> - The authors explicitly acknowledge in their limitations: "with larger compute budgets, one could reuse portions or all of the dataset for multiple epochs of training". Their experimental design completely avoids the complexity of multi-epoch training, which is precisely the core problem our paper addresses as we expilictly incorporate the computaional budget into our formulation.
>
> **Our Work's Coverage:**
>
> - We propose a general CADS framework applicable to various machine learning tasks (vision, language, etc.)
> - We explicitly handle data selection optimization under multi-epoch training scenarios
> - We systematically address the problem that [67] acknowledged but did not handle through our bilevel optimization framework
>
> ### **[Complementary Technical Contributions]**
>
> The two works are actually complementary:
>
> - [67] validates the effectiveness of simple methods in single-epoch training scenarios
> - Our work addresses the multi-epoch training scenarios they left unaddressed but acknowledged as important
> - Our CADS framework can effectively utilize multi-epoch training when computational budget permits, which is an extension of [67]

---

> > ### Comment · Reviewer_vqnc · 2025-08-03
> > **Response to Authors**
> >
> > I thank the authors for answering my questions. Some of my concerns still stand and hence I maintain my rating.
> >
> > 1. The performance of CADS-E is not particularly different from simple approaches such as forgetting score.
> > 2. The computational complexity of CADS is much higher than non-bilevel approaches with only marginal gains (especially in the example level selection). Moreover, approaches such as Forgetting were proposed for example level selection and it is not clear how authors adapted it to source-level selection.

---

> > > ### Author Response · Authors · 2025-08-04
> > > **Part 1: Addressing Reviewer vqnc's Feedback**
> > >
> > > Dear Reviewer vqnc,
> > >
> > > Thank you for your continued feedback. We greatly appreciate your insights, which have helped us improve our paper, and would like to clarify several key aspects of our work and experimental results:
> > >
> > > ### **Q1: Performance of CADS-E compared to Forgetting Score**
> > >
> > > For your concerns, we would like to clarify the following facts:
> > >
> > > - **The marginal gain of CADS-E relative to Forgetting Score mainly comes from the influence of the same pretained model (from `torchvision.models.resnet50`).** This reliance **narrows the performance differences** among various selection methods, as the knowledge embedded in the pretrained models diminishes the gap between them. To validate this, we present results from training from scratch in the section below.
> > > - **Our method demonstrates significant superiority over Forgetting Score in coreset selection problems with noisy labels.** We support this assertion with results from experiments conducted with label noise (see Tab-1.2), where a more pronounced performance gap is observed compared to clean data results. We find that Forgetting Score tends to select data with a higher frequency of flipping between correct and incorrect classifications during training. However, **data with incorrect labels near the classification hyperplane is likely to exhibit this property as well**. Below, we present the noise ratio in the selected coreset, and we will visualize these results in the revised version if accepted.
> > >
> > >
> > > The empirical verifications for the above two facts are given below:
> > >
> > > **[Experiments Trained from Scratch and Noise Data]**
> > >
> > > We expanded the experimental scale and reorganized experiments trained from scratch. The results are shown in the table below, demonstrating that CADS-E significantly outperforms the Forgetting method across various budget sizes.
> > >
> > > #### **Tab-1.1: Accuracy of Training from Scratch**
> > >
> > > | Method     | Computational Budget: 500K | Computational Budget: 1000K | Computational Budget: 1500K | Computational Budget: 2000K | Final Subset Size for Different Budgets |
> > > | ---------- | -------------------------- | --------------------------- | --------------------------- | --------------------------- | --------------------------------------- |
> > > | Forgetting | 22.66%                     | 37.61%                      | 44.22%                      | 46.36%                      | [10K,10K,10K,10K]                       |
> > > | CADS-E     | 25.58%                     | 45.94%                      | 57.65%                      | 62.45%                      | [8276, 12841, 14928, 16435]             |
> > >
> > > Furthermore, we conducted experiments with added noise data to evaluate the robustness of our methods. The results are summarized in the table below, which shows the performance of CADS-E and Forgetting methods alongside the percentage of noise in the selected subset.
> > >
> > > #### **Tab-1.2: Accuracy of Training from Scratch with Noise Data**
> > >
> > > | Method     | Computational Budget: 500K | Computational Budget: 1000K | Computational Budget: 1500K | Computational Budget: 2000K | Selected noise ratio | Final Subset Size for Different Budgets |
> > > | ---------- | -------------------------- | --------------------------- | --------------------------- | --------------------------- | -------------------- | --------------------------------------- |
> > > | Forgetting | 17.53%                     | 33.53%                      | 38.97%                      | 41.39%                      | 8.04%                | [10K,10K,10K,10K]                       |
> > > | CADS-E     | 25.20%                     | 45.58%                      | 56.08%                      | 61.25%                      | 1.16%                | [7655, 11300, 13756, 16199]             |

---

> > > > ### Author Response · Authors · 2025-08-04
> > > > **Part 2: Addressing Reviewer vqnc's Feedback (Continued)**
> > > >
> > > > ### **Q2.1: The computational complexity of CADS**
> > > >
> > > > For the computational complexity, we argue that:
> > > >
> > > > - **[Superiority over non-bilevel approaches.]** The performance in Q1 demonstrates the superiority of our method over non-bilevel approaches, such as Forgetting Score.
> > > > - **[Our contribution to improving the scalability of the bilevel optimization based coreset selection.]** We emphasize that bilevel optimization methods are theoretically known for superior performance, but practical applications are often constrained by their complexity, making large-scale implementation challenging. Our CADS method improves the scalability of these methods, enabling effective data selection within manageable computational costs.
> > > > - **[The computational complexity of our method can be amortized by its repeated use in downstream training.]** Therefore, the efficiency of selecting this coreset is **not the primary concern of this task.** As long as the training complexity remains within **an acceptable range**, we primarily focus on accuracy performance over training time.
> > > >
> > > >
> > > > ### **Q2.2: How we apply Forgetting Score to source-level**
> > > >
> > > > In our initial rebuttal, we mentioned that
> > > >
> > > > > For baseline methods, we combined all 5 groups into a single dataset and applied their selection algorithms to choose 20% of the total data as the training subset." When using the Forgetting method, we treated different sources as a large dataset without differentiating among them.
> > > >
> > > > To further explore the application of Forgetting at the source level, we additionally implemented a variant called **Forgetting***. In this method, we calculated the total forgetting score for samples within each source. Weighted on these scores, we decided how many samples to take from each source, ensuring that the total sample count remains at 20% of the overall dataset. This allows us to apply the forgetting mechanism more precisely while still adhering to the desired budget.
> > > >
> > > > | Method      | Budget: 10 epochs | Budget: 20 epochs | Budget: 30 epochs | Budget: 40 epochs |
> > > > | ----------- | ----------------- | ----------------- | ----------------- | ----------------- |
> > > > | Forgetting  | 22.18%            | 32.87%            | 39.22%            | 42.36%            |
> > > > | Forgetting* | 20.15 %           | 31.45 %           | 36.10 %           | 39.90 %           |
> > > > | **CADS-S**  | **31.20%**        | **42.14%**        | **50.58%**        | **56.13%**        |
> > > >
> > > > We hope this additional information addresses your concerns. If you have further questions or need any clarifications, please feel free to let us know.
> > > >
> > > > Thank you for your time and consideration!

---

> > > > > ### Comment · Reviewer_vqnc · 2025-08-05
> > > > >
> > > > > I thank the authors for the detailed response. My concerns have been addressed. I maintain my initial score.

---

> > > > > > ### Author Response · Authors · 2025-08-05
> > > > > >
> > > > > > Dear Reviewer vqnc,
> > > > > >
> > > > > > Thank you for your continued feedback and for recognizing our efforts in addressing your questions. Your engagement with our work is greatly appreciated.
> > > > > >
> > > > > > We are glad to hear that we addressed your concerns. We will incorporate the results and discussions from the rebuttal process into the final version of our submission.
> > > > > >
> > > > > > Thank you once again for your review.

---

### Official Review · Reviewer_XuGb · 2025-07-03

**Clarity:** 2
**Significance:** 3
**Originality:** 3
**Rating:** 4
**Confidence:** 3

**Summary:**

This paper demonstrates that computational budget constraints significantly impact data selection. The authors propose Computational budget-Aware Data Selection (CADS), formulating it as a bilevel optimization framework. To address two key challenges, they introduce a probabilistic reparameterization strategy and transform the inner optimization into a penalty term within the outer objective. Experimental results and analyses appropriately support the claims.

**Questions:**

See above weakness.

**Ethical Concerns:**

["NO or VERY MINOR ethics concerns only"]

**Final Justification:**

The rebuttal has addressed most of my concerns regarding dataset generation details, budget phase definition, experimental comparison on ImageNet, and the scalability of CADS to other bilevel problems. However, the relevance between the frequency favoritism of DNNs and the proposed method remains unclear. Based on this, I will maintain my rating as borderline accept.

**Limitations:**

Yes

**Quality:**

3

**Strengths And Weaknesses:**

## Strengths

1. This paper reveals an interesting and novel perspective on data selection.

2. This paper elegantly formalizes the problem as a bilevel optimization problem.

3. The proposed CADS is well-motivated, and its effectiveness is thoroughly demonstrated.

## Weaknesses

1. In Sec. 3, how to generate low-/high-frequency datasets? How to define low-/high-budget phases and the difference between them?

2. It seems that there is no direct relevance between the frequency favoritism of deep neural networks and the proposed method. Authors need to further explain the relationship and provide more evidence to support this motivation.

3. One major concern is the lack of comparison between the proposed CADS and established baselines regarding performance and computational cost on widely-used datasets (e.g., ImageNet) and general-purpose models.

4. It seems that CADS is proposed in a general manner. Authors need to discuss the scalability of CADS to other bilevel optimization problems.

---

> ### Author Rebuttal · Authors · 2025-07-30
>
> **We sincerely thank Reviewer XuGb for the constructive feedback.** Below we provide point-by-point responses:
>
> ---
>
> ### **W1: In Sec. 3, how to generate low-/high-frequency datasets? How to define low-/high-budget phases and the difference between them?**
>
> ### **[How to generate dataset]**
>
> As we described in Sec. 3, inspired by [50], the data (x,y) is sampled from the function $y = x + sin(\pi x)/\pi x$. The low-frequency dataset samples data exclusively at $sin(\pi x) = 0$ points, capturing primarily the linear component y = x with additive Gaussian noise. The high-frequency dataset samples uniformly across the domain, preserving the full spectral complexity including oscillatory patterns from the $sin(\pi x)/\pi x$ term.
>
> The **low-frequency** and **high-frequency** datasets in Sec. 3 represent **relative concepts** rather than strict mathematical definitions. This exploratory experiment was designed as a **proof-of-concept** to demonstrate that data preferences can shift with computational budget, which motivated our main contribution.
>
> ###  **[Budget Phase Definition]**
>
> This is simply a name we gave for easy understanding, not a rigorous definition. The distinction between **low-budget** and **high-budget phases** is determined empirically based on the intersection point of the validation loss curves in Fig. 1. The low-budget phase (0-60 steps) is where low-frequency data achieves better performance, while the high-budget phase (80+ steps) is where high-frequency data becomes superior, with a transition point around 60-80 steps.
>
> ---
>
> ### **W2: Relevance between the frequency favoritism of deep neural networks and the proposed method.**
>
> Thank you for this question. We clarify several important points:
>
> - **[Motivation extends beyond frequency favoritism]** While Sec. 3 uses spectral bias as an illustration, our core argument is broader - **Models prefer different types of data during different stages of training**. Spectral bias is one well-documented instance that supports our argument. We argue that different data samples contribute knowledge of varying complexity and learning difficulty, which can be observed in various domains, such as vision, language tasks, etc. Given different budgets, we can expect the optimal coresets to be different.
>
> - **[Sec. 3 as pedagogical illustration]** The spectral analysis serves as illustrative case study to explain underlying concepts, not sole theoretical foundation. CADS framework is agnostic to specific bias types and adapts to different domains' learning patterns.
>
> ---
>
> ### **W3: Comparison with established baselines regarding performance and computational cost on widely-used datasets (e.g., ImageNet) and general-purpose models.**
>
> We conducted comprehensive ImageNet experiments to address this concern.
>
> **Additionally**, in our main paper, we also performed experiments on DomainNet, one of the largest multi-domain datasets, comprising approximately half the scale of ImageNet.
>
> ### **[CADS-E Experiments on ImageNet]**
>
> Due to computational constraints during the rebuttal period, we conducted experiments on ImageNet-100  with 10K samples using ResNet-50. We will include more results in the revised version.
>
> | Method      | Budget: 50K | Budget: 100K | Budget: 200K | Budget: 400K | Final Subset Size         |
> | ----------- | ----------- | ------------ | ------------ | ------------ | ------------------------- |
> | Forgetting  | 81.33%      | 82.39%       | 84.71%       | 86.77%       | [5000,5000,5000,5000]     |
> | EL2N        | 79.18%      | 80.79%       | 83.28%       | 84.82%       | -                         |
> | GraNd-Score | 78.35%      | 79.44%       | 82.10%       | 84.08%       | -                         |
> | Moderate    | 80.61%      | 81.70%       | 84.59%       | 86.84%       | -                         |
> | **CADS-E**  | **82.81%**  | **83.78%**   | **86.46%**   | **88.13%**   | **[4088,6320,7171,8653]** |
>
> CADS-E consistently outperforms all baselines across different computational budgets.
>
> ### **[CADS-S Experiments on ImageNet]**
>
> For CADS-S evaluation, ImageNet-1K partitioned into 5 groups with different label noise levels (0%, 2.5%, 5%, 7.5%, 10%). Experiments were conducted with computational budgets measured in epochs (full dataset traversals). For baseline methods, we combined all 5 groups into a single dataset and applied their selection algorithms to choose 20% of the total data as the training subset.
>
> | Method    | Budget: 10 epochs     | Budget: 20 epochs  | Budget: 30 epochs    | Budget: 40 epochs   |
> | -------------------------- | ------------------------------ | ------------------------------ | ------------------------------ | ------------------------------ |
> | Forgetting                 | 22.18%       | 32.87%           | 39.22%   | 42.36%     |
> | EL2N                       | 20.82%        | 31.35%             | 38.95%   | 41.61%    |
> | GraNd-Score                | 22.47%       | 32.61%            | 39.05%   | 42.79%    |
> | Moderate                   | 26.31%      | 35.06%   | 44.54%    | 49.89%    |
> | **CADS-S**                 | **31.20%**      | **42.14%**   | **50.58%**      | **56.13%**    |
> | **CADS Sampling Ratios r** | [0.85, 0.12, 0.03, 0.00, 0.00] | [0.88, 0.10, 0.02, 0.00, 0.00] | [0.92, 0.06, 0.06, 0.01, 0.00] | [0.95, 0.12, 0.02, 0.02, 0.00] |
>
> ### **[Computational Cost Analysis of Established Baselines]**
>
> We establish unified framework with key parameters:
>
> - **C**: Total compute budget (training cost on full dataset)
> - **N**: Total dataset samples
> - **T**: Training epochs, where T = C/N
>
> All computational overheads are measured relative to the baseline cost **C** of training on the full dataset.
>
> **Note**: For methods requiring pre-trained models, we assume the computational cost of obtaining the necessary pre-trained model is **0.2C** (20% of the full budget), which represents a reasonable approximation for early-stage model training needed to compute gradients or other sample-wise statistics.
>
> | **Method**    | **Total Computational Overhead**    |
> | -------------------------------------- | ------------------------------------------------------------ |
> | **Random Selection**    | **C** (no additional cost)    |
> | **Forgetting Scores**   | **1.2C** (0.2C for pre-training and forgetting computation + C for final training) |
> | **GraNd**         | **1.2C** (0.2C for pre-training and gradient computation + C for final training) |
> | **EL2N**        | **1.2C** (0.2C for pre-training and error computation + C for final training) |
> | **CADS (Ours)**       | **(5/A + 2$\gamma$)C** where A is amortization factor        |
> | **Influence Functions**       | **2C + O(N²)** (full training + expensive Hessian computations + final training) |
> | **Data Shapley**     | **11C~101C** (C for final training + 10C ~ 100C for Monte Carlo approximation with 100~1000 samples) |
> | **Probabilistic Bilevel Optimization** | **50C~100C** (requires 50~100 outer iterations, each involving inner loop model training with computational cost C) |
> | **Greedy Coreset**       | **~($N^2$/T+1)C** where K is coreset size (bilevel solving for each added sample) |
>
> **Summary:** CADS achieves superior efficiency among bilevel methods with overhead **(5/A + 2$\gamma$)C**. With A=5 amortization, total cost reduces to **2$\gamma$C**, $\gamma < 1$.
>
> Traditional methods like Forgetting Scores offer moderate 1.2C cost but lack budget-aware optimization and deliver inferior effectiveness (e.g., lower accuracy in the **Tables of W3**).
>
> Advanced bilevel methods become prohibitively expensive **(50C+)**, while CADS provides optimal balance between computational efficiency and budget-aware capabilities.
>
> **Note:** The core contribution of our work is introducing computational budget constraints into coreset selection, rather than optimizing selection time alone. Once learned, **a coreset can be reused multiple times** across different training scenarios, model architectures, and experiments. Therefore, the computational investment in coreset selection can be amortized over multiple uses, making the selection efficiency a secondary consideration compared to the quality of the resulting coreset under budget constraints.
>
> ---
>
> ### **W4: Discuss the scalability of CADS to other bilevel optimization problems.**
>
> We actually have a successful example currently under submission that applies  a CADS-like method to **diffusion models**, since DDPM convergence is extremely resource-intensive.
>
> ### **[Applicability Conditions]**
>
> CADS is specifically designed to address a fundamental challenge in bilevel optimization: **scenarios where the inner-level problem cannot converge due to computational budget constraints**. The key insight of our approach is that many practical bilevel problems suffer from this issue, where traditional methods assume unlimited computational resources for inner-level convergence—an assumption that rarely holds in real-world applications.
>
> ### **[Loss Estimation Accuracy]**
>
> The scalability of CADS to other bilevel problems primarily depends on the **accuracy of loss estimation** in the target domain. As demonstrated in our response to reviewer **cbM7** regarding loss estimation generalization, our proposed loss estimator shows relatively stable predictive performance across the experimental domains in this work.
>
> ### **[Validation Framework for New Domains]**
>
> For applying CADS to other bilevel optimization problems, we propose the following validation approach:
>
> 1. **Loss Estimator Generalization Assessment**: Conduct experiments similar to our generalization evaluation to test whether the data distribution characteristics in the new domain can be reliably predicted by a loss estimator.
>
> 2. **Domain-Adaptive Estimator Design**: While our current estimator works well for data selection problems, different bilevel scenarios may benefit from **specialized estimators tailored to their specific requirements** and data characteristics.

---

> > ### Comment · Reviewer_XuGb · 2025-08-04
> >
> > Thanks for the rebuttal. Authors have addressed most of my concerns, and I will maintain my rating.

---

> > > ### Author Response · Authors · 2025-08-04
> > >
> > > Dear Reviewer XuGb,
> > >
> > > Thank you for your thoughtful and constructive feedback on our paper. We appreciate your insights and are glad to hear that we have addressed most of your concerns.
> > >
> > > If there are any remaining issues that we might have overlooked, we would appreciate the opportunity to discuss these areas with you. We are committed to addressing any outstanding questions to improve our final submission.
> > >
> > > Your comments have been invaluable, and we are grateful for your support of our research.
> > >
> > > Thank you once again!

---

### Official Review · Reviewer_TtXp · 2025-07-06

**Clarity:** 3
**Significance:** 2
**Originality:** 2
**Rating:** 4
**Confidence:** 3

**Summary:**

The paper proposes a two-step algorithm to incorporate the computational budget of a training model in  their data selection procedure. It advocates to keep an eye on the compute budget during data selection and not treat it as a fixed hyper parameter. I like the hypothesis of the paper, but I have some concerns over the overall exposition and validation of the approach.

**Questions:**

- Why the first step needs Hessians computation?
- How is optimality defined in the second step?
- How can one know how much time a model would need to train?
- It is not clear to what the compute budget refers to? Is it the compute budget for the model or the compute budget for the data mining process?
- What computational overhead the algorithm adds on top of the computational cost of training a network?
- Can you provide more validations on generative models showing different metrics, how they get affected by the compute budget?

**Ethical Concerns:**

["NO or VERY MINOR ethics concerns only"]

**Final Justification:**

My major concern (and other concerns) regarding the overhead introduced by this paper is properly addressed.

I think the paper has a strong hypothesis, and after checking the other rebuttals and reviews from other reviewers, I feel that the paper is going to make a good contribution.

I am now borderlin positive and will rely on fellow reviewers to decide whether this paper is above NeurIPS bar or not.
Why borderline, is due to the heavy overhead of the proposed approach. But I don't think that is a major bottleneck and future research can develop better strategies to amortize this cost.

**Limitations:**

yes

**Paper Formatting Concerns:**

The paper is fairly well written.

**Quality:**

2

**Strengths And Weaknesses:**

Strengths:
- The paper raises a very important aspect of the training process. Computation budget should be part of the training process and the data selection
- Some initial results are shown on the MNIST dataset, and some other datasets to validate the hypothesis.


Weakness:
- The text did not make it clear what does compute budget refers to
- The two-step algorithm in the introduction misses the context, like why Hessians are needed in the first step, what optimality refers to in the data selection.
- The paper seems to be motivated based on the observation of spectral bias from Rahman et al. This is insightful but does that generalizes to different problems in machine learning?
- Solid validation is missing, for example, how the generalization of a network gets affected with the compute budget (in terms of FID scores).

Overall I think the idea is not sufficiently validated.

---

> ### Author Rebuttal · Authors · 2025-07-30
>
> **We sincerely thank Reviewer TtXp for the constructive feedback.** Below we provide point-by-point responses:
>
> ---
>
> ### **W1&Q4: Is it the compute budget for  model training or data mining process?**
>
> **The compute budget C refers specifically to model training, not data mining process.** It is defined in Section 4.1 (lines 175-181). To be precise, our method is formalized as:
>
> $\min \mathcal{L}_{val}(\theta_C(m)) \quad \text{s.t.} \quad \theta_C(m) = \text{Train}(m,C)$
>
> where $C$ represents the computational cost for training the model on the given subsets, which **equals to #iteration*batch-size**.
>
>
> ---
>
> ### **W2&Q1&Q2: Why Hessians are needed in the first step, what optimality refers to in the data selection.**
>
> We think your concern is based on our statement in lines 61 to 62.
>
> ### **[Why Hessians are Required]**
> The Hessian requirement in bilevel optimization is extensively studied in prior work [r1-3]. Our bilevel formulation follows the standard structure:
>
> - **Inner**: $\theta^{\*}(m) = \arg \min_{\theta} \mathcal{L}_{\text{train}}(\theta, m)$
> - **Outer**: $\min_{m} \mathcal{L}_{\text{val}}(\theta^{\*}(m))$
>
> To optimize the outer loop, we compute $\nabla_{m} \mathcal{L}_{\text{val}}(\theta^{*}(m))$ using chain rule:
>
> $$\nabla\_{m} \mathcal{L}\_{\text{val}}(\theta^{\*}(m)) = \nabla\_{\theta} \mathcal{L}\_{\text{val}} \cdot \nabla\_{m} \theta^{\*}(m)$$
>
> Since $\theta^{\*}(m)$ satisfies $\nabla\_{\theta} \mathcal{L}\_{\text{train}}(\theta^{\*}(m), m) = 0$, implicit differentiation calculates $\nabla_{m} \theta^{\*}(m)$ as :
>
> $\nabla_m \nabla\_{\theta} \mathcal{L}\_{\text{train}}(\theta^{\*}(m), m)=0$,
>
> i.e.,
>
> $\nabla^2\_{\theta \theta} \mathcal{L}\_{\text{train}}(\theta^{\*}(m), m)^\top \nabla_{m} \theta^{\*}(m) + \nabla^2\_{\theta m} \mathcal{L}\_{\text{train}} (\theta^{\*}(m), m)=0.$
>
> Therefore,
>
> $$\nabla_{m} \theta^{\*}(m) = -\left[\nabla^{2}_{\theta\theta} \mathcal{L}\_{\text{train}}\right]^{-1} \cdot \nabla^{2}\_{\theta m} \mathcal{L}\_{\text{train}}$$
>
> The term $\nabla^{2}\_{\theta\theta} \mathcal{L}\_{\text{train}}\in R^{d\times d}$ is the **Hessian matrix**, making Hessian computation almost unavoidable in exact bilevel optimization as  the dimension d of $\theta$ is always larger than 1M.
>
> ### **[Definition of Optimality]**
>
> "Optimality" refers to standard bilevel definition: For each iteration in the outer loop, inner loop needs to find the optimal solution $\theta^{*}(m)$ for  $\min_{\theta} \mathcal{L}_{\text{train}}(\theta, m)$. This can lead to a huge cost since we need to perform lots of outer loop iteration for solving the bilevel problem.
>
>
> We will include these details in the revised version if accepted.
>
> [r1] J. Domke. Generic methods for optimization-based modeling. AISTATS, 2012.
>
> [r2] F. Pedregosa. Hyperparameter optimization with approximate gradient. ICML, 2016.
>
> [r3] T. Chen et al. A single-timescale method for stochastic bilevel optimization. AISTATS, 2021.
>
> ---
>
> ### **W3: The paper seems to be motivated based on the observation of spectral bias from Rahman et al. This is insightful but does that generalizes to different problems in machine learning?**
>
> Thank you for this question. We clarify several important points:
>
> - **[Spectral bias is universal.]** This represents a fundamental property extensively studied across ML community [r4,5]. Neural networks consistently learn low-frequency before high-frequency components across various architectures and domains.
>
> - **[Motivation extends beyond spectral bias.]** While Sec. 3 uses spectral bias as an illustration, our core argument is broader - **Models prefer different types of data during different stages of training**. Spectral bias is one well-documented instance that supports our argument. We argue that different data samples contribute knowledge of varying complexity and learning difficulty, which can be observed in various domains, such as vision, language tasks, etc. Given different budgets, we can expect the optimal coresets to be different.
>
> - **[Sec. 3 as pedagogical illustration.]** The spectral analysis serves as illustrative case study to explain underlying concepts, not sole theoretical foundation. CADS framework is agnostic to specific bias types and adapts to different domains' learning patterns.
>
> We appreciate this question as it allows us to clarify that our contribution addresses a fundamental aspect of neural network training that extends well beyond any single type of bias.
>
> [r4] N. Rahaman et al. On the spectral bias of neural networks. ICML, 2018.
>
> [r5] Z.-Q. J. Xu et al. Frequency principle: Fourier analysis sheds light on deep neural networks. arXiv, 2019.
>
> ---
>
> ### **W4&Q6: How the generalization of a network gets affected with the compute budget (in terms of FID scores).**
>
> We conducted additional experiments on generative models to demonstrate CADS's effectiveness across computational budgets.
>
> ### **[Exp 1: CADS-E on StyleGAN2]**
>
> StyleGAN2 on CIFAR-10 with varying budgets. Random baseline used fixed 25600 samples.
>
> | Budget (img) | Random FID$\downarrow$ | CADS-E  FID$\downarrow$ | CADS Subset Size |
> | ------------ | ---------------------- | ----------------------- | ---------------- |
> | 1000k        | 34.73                  | **31.47**               | 26304            |
> | 2000k        | 17.04                  | **15.39**               | 29518            |
> | 3000k        | 12.40                  | **11.45**               | 32881            |
> | 4000k        | 9.51                   | **8.75**                | 38091            |
>
> ### **[Exp 2: CADS-S on DDPM]**
>
> DDPM (EDM implementation) on CIFAR-10 with 5 noise groups (σ = 0.00, 0.01, 0.02, 0.03, 0.04). Uniform baseline uses r = [0.2, 0.2, 0.2, 0.2, 0.2]. Each group contains 10,000 images.
>
> | Budget (img) | Uniform FID$\downarrow$ | CADS-S FID$\downarrow$ | CADS Ratio r            |
> | ------------ | ----------------------- | ---------------------- | ----------------------- |
> | 2000k        | 49.45                   | **26.55**              | [0.97,0.08,0.01,0,0]    |
> | 4000k        | 41.69                   | **17.15**              | [1.00,0.15,0.08,0,0]    |
> | 8000k        | 35.31                   | **8.33**               | [1.00,0.12,0.05,0.02,0] |
> | 16000k       | 30.94                   | **4.09**               | [1.00,0.23,0.03,0.01,0] |
>
> Results validate that computational budget considerations lead to measurably better generative model performance.
>
> ---
>
> ### **Q3: How can one know how much time a model would need to train?**
>
> The computational budget is typically determined by **task-specific constraints** rather than arbitrary choices.
>
> **Determining Training Time Budget**
>
> Budget comes from operational requirements: news recommendation systems in the industry may need updates periodically, e.g., every 20 minutes (leaving 5-10 minutes for training), cloud budgets ($10/run) convert to time via pricing.
>
> Users estimate budgets through pilot experiments, hardware profiling, or converting monetary limits to compute time.
>
> **CADS Framework Flexibility**
>
> Our method treats budget as user-defined parameter and optimizes data selection under this constraint. Users start with rough estimates and refine through ablation studies. CADS explicitly incorporates practical constraints into optimization, unlike traditional methods ignoring computational limits.
>
> Budget reflects real-world operational needs, making our approach highly relevant for production systems.
>
> ---
>
> ### **Q5: What computational overhead the algorithm adds on top of the computational cost of training a network?**
>
> Thank you for this important question. Our CADS algorithm's overhead consists of two components:
>
> **Linear Loss Estimator Overhead:**
> To construct loss surrogate function l(|m|) (Section 4.3), we sample m different subset sizes. Using m=5 subset sizes, the overhead is mC where C is total compute budget. Since this function can be used in each run, this cost can be amortized across multiple runs with factor A, i.e., A is the number of runs, giving effective overhead mC/A.
>
> **Core Algorithm Overhead:**
> Algorithm performs M iterations with K=2 Monte Carlo samples each. We let the subset sized of $\gamma$N, where $\gamma < 1$, resulting in computational cost $\gamma$MKC/T, where T=C/N represents epochs in conventional training.
>
> **Total Overhead Analysis:**
> Total overhead = mC/A + $\gamma$MKC/T. With practical settings (m=5, K=2, M≈T), this becomes:
>
> **Total Overhead = 5C/A + 2$\gamma$C = (5/A + 2$\gamma$)C**
>
> The additional overhead over baseline training cost C is **(5/A + 2$\gamma$ - 1)C**.
>
> **Summary:**
>
> 1. When M≈T, the total overhead is  (5/A + 2$\gamma$)C
> 2. Overhead significantly reduced by increasing amortization factor A through reusing estimators across multiple selection tasks
> 3. With A=5 reuses, the additional overhead reduces to just 2$\gamma$C
>
> **Note:** The core contribution of our work is introducing computational budget constraints into coreset selection, rather than optimizing selection time alone. Once learned, **a coreset can be reused multiple times** across different training scenarios, model architectures, and experiments. Therefore, the computational investment in coreset selection can be amortized over multiple uses, making the selection efficiency a secondary consideration compared to the quality of the resulting coreset under budget constraints.

---

> ### Comment · Reviewer_TtXp · 2025-08-03
>
> Thanks for addressing my concerns. My major concern regarding the overhead introduced by this paper is properly addressed. I think this is significant and should be highlighted in the main paper. This can help design the optimization strategy carefully.
>
> Please make sure to add all the promised results and discussions from the rebuttal in the final version of the paper.

---

> > ### Author Response · Authors · 2025-08-04
> >
> > Dear Reviewer TtXp,
> >
> > Thank you for your thoughtful and constructive feedback on our rebuttal. We appreciate your recognition of our efforts to clarify the overhead introduced by our algorithm. We will ensure that the discussions from our rebuttal, particularly regarding the algorithm overhead, are incorporated into the final manuscript.
> >
> > We are open to discussing any further questions you may have. Additionally, if possible, could you share your final rating? We’re unable to see updates on the ratings due to recent policy changes, and your input would be greatly appreciated. If that's inconvenient, we completely understand.
> >
> > Thank you once again for your valuable feedback.

---

### Comment · Area_Chair_2u5w · 2025-08-04
**Discussions**

Dear Reviewers,

Thank you very much for your time and efforts. As we are approaching the deadline, we kindly ask you to review the rebuttal and share any remaining concerns with the authors for discussion.

Best regards,

 AC

---

### Note · Authors · 2025-08-13

Dear PC, SAC, AC and Reviewers,

We sincerely appreciate the valuable feedback and the constructive discussion with all reviewers. We are grateful for the **positive feedback** from all reviewers on our work and rebuttal. We were encouraged to find **that our rebuttal addressed their concerns without raising new issues.** We confirm that all promised revisions and additional results will be incorporated into the final version.

For the convenience of the committee's final review, we would like to briefly summarize what we believe are the key contributions of our work, supported by the reviewers' positive feedback:

- **Proposing a Novel, and Important Aspect of Data Selection**: Existing methods often overlook computational budgets. Our work highlights that **the computational budget should be an integral part of the data selection strategy**, as different budgets entail distinct data requirements. To this end, **we propose CADS, a bilevel optimization framework**, and we hope it may inspire the development of more budget-aware algorithms.
- **Addressing Non-Convergence in Budget-Constrained Bilevel Optimization**: A key challenge is that the inner-level optimization in our problem cannot reach full convergence due to the budget constraint, violating standard assumptions of bilevel optimization. To overcome this, we employ a policy gradient approach along with a loss estimator to **bypass the reliance on inner-level optimality**, offering a promising path for similar bilevel problems where inner-level optimality is not guaranteed.
- **Enhancing the Efficiency of Bilevel Optimization**: To improve the practicality of our method for larger tasks, we introduce a penalty term constraint that approximates the inner-level loop. This modification results in significant efficiency improvements, retaining the precision of the bilevel approach while enhancing its scalability for larger-scale applications.

We are very grateful for the insightful feedback, which will be invaluable for our future research. We hope our work can contribute to the discussion and exploration within our community.

Sincerely,

Authors

---

### Decision · Program_Chairs · 2025-09-17

**Decision:**

Accept (poster)

**Comment:**

Overall, this paper makes a timely and significant contribution by establishing computational budget as a core principle in data selection and introducing an effective algorithmic framework (CADS) to operationalize this idea. The methodological innovations are well-motivated, the empirical results are compelling, and the work has important implications for large-scale model training under practical resource constraints. It is essential that the authors address the reviewers’ concerns in the final version.